# ONEFLOWSEQ: ACHIEVING ONE-STEP GENERATION FOR DIFFUSION LANGUAGE MODELS VIA LIGHTWEIGHT DISTILLATION

## ABSTRACT

Autoregressive models dominate Seq2Seq generation but suffer from slow, error-prone token-by-token decoding. Diffusion language models (DLMs) enable parallel refinement and global coherence, yet their iterative denoising requires hundreds of steps, limiting practicality. Attempting to address this issue, we propose **One-FlowSeq**, a novel framework that distills a powerful multi-step diffusion teacher (LLaDA-8B-Instruct) into a one-step generator via MeanFlow-based supervision and parameter-efficient prompt tuning. Our OneFlowSeq introduces a Jacobian-vector product signal that provides richer guidance than conventional distillation, allowing the student model to not only match the 128-step teacher model[1] in terms of one-step generation quality. Experiments on paraphrasing, text simplification, and question generation benchmarks show that our OneFlowSeq achieves state-of-the-art performance, while reducing trainable parameters by $1600\times$ and delivering inference speeds orders of magnitude faster than both autoregressive and multi-step diffusion baselines. This work establishes one-step diffusion as a practical and scalable paradigm for Seq2Seq generation.

## 1 INTRODUCTION

Autoregressive (AR) language models (Brown et al., 2020; Radford et al., 2019; Touvron et al., 2023) have long dominated sequence-to-sequence (Seq2Seq) (Sutskever et al., 2014; Yousuf et al., 2020) tasks in natural language processing. While their token-by-token generation ensures consistency, it introduces fundamental bottlenecks: inference latency scales linearly with sequence length, and unidirectional context hinders global planning. As an emerging paradigm, diffusion language models (DLMs) (Ye et al., 2025; Li et al., 2025; Zhang et al., 2025) promise to overcome these issues. Their parallel generation and holistic refinement mechanisms naturally support bidirectional reasoning and global coherence. However, their practical application is hindered by a significant drawback: the iterative denoising process often requires hundreds or thousands of steps, making inference prohibitively slow.

This tension has sparked intense interest in one-step diffusion (Shen et al., 2025; Xu et al., 2025; Song & Dhariwal, 2023; Frans et al., 2025). Two recent lines of work have laid the groundwork. MeanFlow (Geng et al., 2025) offers a principled reformulation of diffusion dynamics via average velocity, enabling stable one-step generation without distillation. However, its prohibitive cost of training large models from scratch makes it impractical for the language domain. In contrast, DLM-One (Chen et al., 2025) demonstrates that score distillation can compress multi-step language diffusion into a single forward pass, achieving huge inference acceleration. Yet, it does so by re-training billions of parameters and relying on adversarial stabilization, which shifts the burden from inference to training inefficiency. This creates a deadlock: one-step diffusion is either elegant but prohibitively expensive to train, or fast at inference but bloated in training cost. Neither provides a viable path toward scalable, resource-friendly Seq2Seq generation.

Attempting to tackle this problem, we introduce the OneSeqFlow framework, which combines the stability of MeanFlow with the practicality of distillation in a new way, i.e., instead of rebuilding or

---

[1]While the original LLaDA paper employs variable sampling steps for different datasets in its official evaluation, for a fair comparison, we use a fixed 128 steps for all experiments in this work.

retraining full models, we freeze a large multi-step teacher model (LLaDA-8B-Instruct) (Nie et al., 2025) and distill its dynamics into a tiny soft prompt module (Li & Liang, 2021) with only ~5M trainable parameters. The core process is a Jacobian-vector product (JVP) supervision signal, which encodes not just the first-order direction but also the second-order dynamics of the teacher. This richer signal provides the student model with a clearer learning objective, allowing it to reach the level of its teacher model in generation quality in just one step, which we consistently observe across benchmarks. Our OneSeqFlow achieves performance equivalent to the teacher model on paraphrase, text simplification, and question generation tasks while reducing the number of trainable parameters by 1600 $\times$. At inference, it eliminates the classic trade-off between speed and quality: a single-step OneSeqFlow forward pass runs orders of magnitude faster than both AR baselines and multi-step diffusion, without sacrificing diversity or semantic fidelity, and achieves extremely fast speed and advanced results on multiple Seq2Seq tasks. Hence, our OneSeqFlow transforms one-step diffusion from a theoretical curiosity into a scalable, deployable paradigm for Seq2Seq generation.

## 2 BACKGROUND

Our work introduces a novel distillation framework that transforms a large, multi-step diffusion language model into a highly efficient one-step generator. We build upon a masked diffusion model, i.e., LLaDA, and design a distillation process that combines the theoretical stability of the MeanFlow method with a parameter-efficient prompt finetuning strategy.

### 2.1 THE LLaDA TEACHER: A DIFFUSION MODEL FOR LLMs

Our teacher model is LLaDA (Large Language Diffusion with mAsking). Unlike conventional autoregressive models, LLaDA is a Masked Diffusion Model (MDM) trained from scratch on 2.3 trillion tokens. Its generative process consists of a forward masking procedure and a learned reverse process. The forward process corrupts a clean sequence $\mathbf{x}_0$ by independently replacing each token with a `[MASK]` token with probability $t \in [0, 1]$, yielding a corrupted sequence $\mathbf{x}_t$. The reverse process is parameterized by a bidirectional Transformer $p_\theta$ that predicts the original tokens from $\mathbf{x}_t$:

$$\mathcal{L}(\theta) = -\mathbb{E}_{t,\mathbf{x}_0,\mathbf{x}_t} \left[ \sum_{i=1}^{L} \mathbb{I}[x_t^i = \text{MASK}] \log p_\theta(x_0^i \mid \mathbf{x}_t) \right]. \quad (1)$$

While LLaDA achieves performance comparable to strong autoregressive LLMs (team, 2024; Yang et al., 2024), its iterative denoising procedure requires multiple steps, incurring significant latency. This motivates exploring a one-step generative mechanism, which we introduce next through the MeanFlow method.

### 2.2 THE MEANFLOW METHOD FOR ONE-STEP GENERATION

Standard flow-matching models (Lipman et al., 2023; Holderrieth & Erives, 2025; Jin et al., 2025) learn a neural network $v_\theta(\mathbf{z}_t, t)$ to estimate the expected instantaneous velocity $\bar{v}_t = \mathbb{E}[v_t \mid \mathbf{z}_t]$ of a flow from noise to data. Sampling involves numerical integration of this velocity field; large discrete steps often cause ambiguity and mode averaging. MeanFlow addresses this limitation by predicting the *average velocity* over an interval $[r, t]$:

$$u(\mathbf{z}_t, r, t) = \frac{1}{t - r} \int_r^t v(\mathbf{z}_\tau, \tau) d\tau. \quad (2)$$

Learning $u$ enables single-step generation. The key is the **MeanFlow Identity**, which relates $u$ to $v$ without explicit integration:

$$u(\mathbf{z}_t, r, t) = v(\mathbf{z}_t, t) - (t - r) \frac{d}{dt} u(\mathbf{z}_t, r, t). \quad (3)$$

This identity yields a principled training objective. The derivative term $\frac{d}{dt}u$ is efficiently obtained via a Jacobian-vector product (JVP), i.e., the product of the Jacobian $\partial u / \partial \mathbf{z}$ with a vector, computable with one forward–backward pass in modern autodiff libraries. While the MeanFlow identity provides strong theoretical foundations, applying it to large models like LLaDA from scratch remains computationally prohibitive. This challenge motivates a distillation-based solution.

## 2.3 DISTILLATION VIA PROMPT FINETUNING

To avoid the cost of full-model training with MeanFlow, we use a Parameter-Efficient Fine-Tuning (PEFT) strategy (Xu et al., 2023). Specifically, we distill the dynamics of the frozen multi-step LLaDA-8B-Instruct teacher into a compact soft prompt module with only ∼5M trainable parameters.

The main challenge is to condition the student model on an interval $(r, t)$ and predict $u_\theta(\mathbf{x}_t, r, t)$, although the frozen LLaDA accepts only a single time input $t$. We address this with a trainable **Soft Prompt Module**. This module employs a lightweight MLP(Popescu et al., 2009), the *Prompt Network*, to map embeddings of $r$ and $t$ into soft prompt vectors. These vectors are injected as prefixes into the Key and Value sequences of every self-attention layer (Vaswani et al., 2023) in the frozen LLaDA. Through prefix-tuning, the prompt influences the full model to condition on $(r, t)$.

During training, only the parameters $\theta$ of the Soft Prompt Module are updated. The objective is:

$$\mathcal{L}_{\text{distill}}(\theta) = \mathbb{E} \left\| u_\theta(\mathbf{x}_t, r, t) - \text{sg}\left( v_{\text{teacher}}(\mathbf{x}_t, t) - (t - r)\frac{d}{dt}u_\theta(\mathbf{x}_t, r, t) \right) \right\|^2, \tag{4}$$

where $\text{sg}(\cdot)$ denotes the stop-gradient operator, preventing gradients from flowing into the teacher. This objective couples instantaneous velocity supervision from the frozen teacher with a second-order signal from the JVP. As a result, the compact prompt module reliably learns a one-step mapping of the teacher's dynamics.

## 3 THE ONEFLOWSEQ FRAMEWORK

To address the challenges of speed and quality in Seq2Seq tasks, we introduce **OneFlowSeq**, a novel framework that distills large, multi-step diffusion language models into highly efficient one-step generators. By combining the theoretical rigor of MeanFlow with a highly resource-efficient distillation strategy, OneFlowSeq achieves a unique combination of performance and scalability.

### 3.1 FRAMEWORK OVERVIEW

At its core, our OneFlowSeq establishes a symbiotic relationship between two key components: (1) a frozen, pre-trained multi-step diffusion model (LLaDA-8B-Instruct) serving as the **teacher**, and (2) the **student model**, which comprises the frozen teacher backbone augmented with a lightweight, trainable Soft Prompt Module. While we optimize only the prompt parameters to minimize training costs, the term 'student' refers to this entire functional unit during inference. The core idea is to distill the complex, iterative dynamics of the teacher into the compact prompt module. Rather than training from scratch, the student learns to predict the *average velocity* of the teacher's denoising trajectory over an interval, guided by the MeanFlow identity. This enables efficient generation while preserving the high performance of the original diffusion model, addressing the inference bottlenecks of multi-step methods. Figure 1 provides a visual overview of the workflow.

### 3.2 ARCHITECTURAL DESIGN

The foundation of our OneFlowSeq is a pre-trained LLaDA-8B-Instruct model, which serves as the teacher model. Its parameters are kept entirely **frozen** throughout the distillation process. This strategic choice dramatically minimizes computational overhead and memory usage, thereby enabling scalability to even larger models. The central architectural challenge is to enable the frozen teacher, which expects a single time input $t$, to operate over a time interval $(r, t)$ as required by MeanFlow. Our solution is a trainable **Soft Prompt Module** comprising a small Multi-Layer Perceptron (MLP), termed the *Prompt Network*, which maps sinusoidal time embeddings of $r$ and $t$ to a sequence of $k$ soft prompt vectors. These vectors are regarded as prefixes to the Key (K) and Value (V) sequences in the self-attention mechanism of **every Transformer layer** in the LLaDA base. This prefix-tuning approach empowers the small module to steer the behavior of the massive base model without modifying any of its core weights, making it exceptionally parameter-efficient.

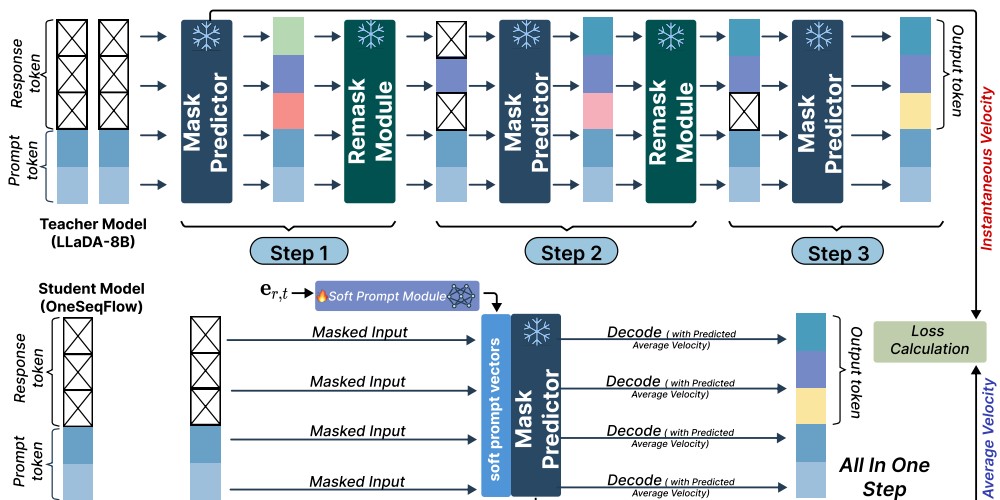

Figure 1: An overview of our OneFlowSeq framework. The frozen teacher model (top) is a multi-step generator that provides an instantaneous velocity target. The student model (bottom) uses a small yet trainable soft prompt module to guide the shared, frozen backbone in predicting the average velocity in a single step. The prompt module is trained exclusively by minimizing a distillation loss between the two velocities, resulting in a highly efficient one-step generator.

### 3.3 TRAINING OBJECTIVE AND PROCEDURE

**MeanFlow preliminaries.** We distinguish between the teacher's instantaneous velocity $v(\mathbf{x}_t, t)$ and the average velocity $u(\mathbf{x}_t, r, t)$ over an interval:

$$u(\mathbf{x}_t, r, t) \;=\; \frac{1}{t-r} \int_r^t v(\mathbf{x}_\tau, \tau) \, d\tau, \quad 0 \le r < t \le 1. \tag{5}$$

Differentiating w.r.t. $t$ yields the **MeanFlow identity**:

$$u(\mathbf{x}_t, r, t) \;=\; v(\mathbf{x}_t, t) \;-\; (t-r)\frac{d}{dt}u(\mathbf{x}_t, r, t), \tag{6}$$

with total derivative

$$\frac{d}{dt}u(\mathbf{x}_t, r, t) \;=\; v(\mathbf{x}_t, t)\, \partial_\mathbf{x} u(\mathbf{x}_t, r, t) \;+\; \partial_t u(\mathbf{x}_t, r, t), \tag{7}$$

which corresponds to a Jacobian-vector product (JVP) along the tangent $(v, 0, 1)$. Boundary and consistency properties hold: (i) $\lim_{r \to t} u(\mathbf{x}_t, r, t) = v(\mathbf{x}_t, t)$; (ii) for any $s \in (r, t)$,

$$(t-r)\, u(\mathbf{x}_t, r, t) \;=\; (s-r)\, u(\mathbf{x}_s, r, s) \;+\; (t-s)\, u(\mathbf{x}_t, s, t). \tag{8}$$

**Representation Space.** To rigorously apply the continuous-time MeanFlow to discrete token sequences, we define all velocity-related operations in a continuous representation space. Specifically, we operate in the **logit space**. Let $\mathbf{z}_t \in \mathbb{R}^{L \times V}$ represent the logit distribution for a sequence of length $L$ over a vocabulary of size $V$ at time $t$. The teacher's instantaneous velocity $v_{\text{teacher}}(\mathbf{x}_t, t)$ and the student's average velocity $u_\theta(\mathbf{x}_t, r, t)$ are both parameterized to take the corrupted token sequence $\mathbf{x}_t$ as input but produce outputs in this logit space, i.e., $\mathbb{R}^{L \times V}$. This ensures that the subtractions and integrations inherent to the MeanFlow identity (Eq. 6) are well-defined vector operations.

*Discrete diffusion note:* Although Eq. equation 6 is derived for continuous state spaces, our discrete formulation operates over a continuous time variable $t$ representing the masking probability. We therefore interpret the velocity field as governing the **expected** denoising trajectory in the space of token probabilities. Under this interpretation, where integrals are replaced by expectations over the masking process, the MeanFlow identity remains a powerful guiding signal for distillation (a rigorous justification is provided in Appendix A).

---

**Algorithm 1** OneFlowSeq Training (*Discrete diffusion via masking corruption*)

---

**Require:** Frozen teacher model $p_{\text{teacher}}$, trainable Soft Prompt Module $p_\theta$

1: **for** each training step **do**
2:      Sample clean data $\mathbf{x}_0$ from dataset
3:      Sample $r \sim \text{Uniform}(0, 0.8)$, $t \sim \text{Uniform}(r + 0.1, 1.0)$
4:      Generate $\mathbf{x}_t$ by masking $\mathbf{x}_0$ with probability $t$         ▷ discrete diffusion corruption
5:      Compute $u_\theta(\mathbf{x}_t, r, t)$
6:      Compute $v_{\text{teacher}}(\mathbf{x}_t, t)$
7:      Compute JVP term $\frac{d}{dt} u_\theta(\mathbf{x}_t, r, t)$ using automatic differentiation
8:      Calculate $\mathcal{L}_{\text{distill}}$ (Eq. 9) and update $\theta$
9: **end for**

---

**Distillation objective.** The training objective transfers the teacher's dynamics to the student using the rich signal from Eq. 6. Instead of approximating the derivative term numerically, we compute it precisely and efficiently using automatic differentiation. The resulting distillation loss is:

$$\mathcal{L}_{\text{distill}}^{\text{disc}}(\theta) \;=\; \mathbb{E}_{\mathbf{x}_0,\, r,\, t} \left\| u_\theta(\mathbf{x}_t, r, t) \;-\; \text{sg}\Big( v_{\text{teacher}}(\mathbf{x}_t, t) \;-\; (t - r)\, \frac{d}{dt} u_\theta(\mathbf{x}_t, r, t) \Big) \right\|^2, \quad (9)$$

where $\text{sg}(\cdot)$ is the stop-gradient operator ensuring gradients flow only to the student's parameters $\theta$. This objective uniquely combines direct supervision from the teacher's instantaneous velocity ($v_{\text{teacher}}$) with a self-consistency signal derived from the student's own dynamics ($\frac{d}{dt} u_\theta$), encouraging it to learn a globally consistent flow field. This richer, second-order signal is critical for high-fidelity distillation, as analyzed from a Sobolev-like perspective in Appendix B.2. It is important to note that this JVP supervision is not merely a regularization term but a critical structural signal. Unlike standard MeanFlow which may struggle in high-dimensional discrete logit spaces, the JVP signal provides second-order guidance that stabilizes the one-step mapping, effectively bridging the gap between continuous flow theory and discrete token generation (see Appendix B.2 for theoretical analysis)

The total derivative term $\frac{d}{dt} u_\theta(\mathbf{x}_t, r, t)$ is implemented as a Jacobian-vector product (JVP), as expanded in Eq. 7. This JVP is computed efficiently using built-in functions from deep learning frameworks (e.g., 'torch.func.jvp'). It calculates the directional derivative of the student network $u_\theta$ with respect to its inputs $(\mathbf{x}_t, r, t)$ along the tangent vector $(v_{\text{teacher}}(\mathbf{x}_t, t), 0, 1)$. Crucially, the entire target for the student model is wrapped in the stop-gradient operator. This means that during backpropagation, the JVP's output is treated as a constant, which prevents costly second-order derivatives and keeps the computational overhead of training minimal.

**Implementation Details.** The Prompt network is a 2-layer MLP with a hidden dimension of 32, which maps the input time embeddings to the final output size of $k \times d$ (where $k = 32$, $d = 4096$). We use the AdamW optimizer with a learning rate $\eta = 5 \times 10^{-4}$, a weight decay of 0.01, and a batch size of 32. The model is trained on 8 NVIDIA A100 GPUs for 80,000 steps. This configuration trains only $\sim$5M parameters and adds negligible computational overhead per forward pass compared to the frozen teacher.

### 3.4 INFERENCE

All inference updates are performed in the logit space. We start with an initial logit tensor $\mathbf{z}_1$, which represents the maximally corrupted sequence (e.g., a zero tensor or logits corresponding to a uniform distribution over the vocabulary).

**Multi-step (K-NFE) Inference.** Given a partition $1 = t_K > t_{K-1} > \cdots > t_0 = 0$, we iteratively update the logit tensor:

$$\mathbf{z}_{t_{i-1}} \;=\; \mathbf{z}_{t_i} \;-\; (t_i - t_{i-1})\, u_\theta\big(\mathbf{x}_{t_i}, t_{i-1}, t_i\big), \quad i = K, \ldots, 1, \quad (10)$$

where $\mathbf{x}_{t_i}$ is the token sequence obtained by decoding the logits $\mathbf{z}_{t_i}$ at each intermediate step. This intermediate decoding can be simplified by directly feeding the continuous embeddings corresponding to $\mathbf{z}_{t_i}$ into the model.

**Single-step (1-NFE) Inference.** As a special case with $K=1$, we set $(r, t) = (0, 1)$ and directly compute the final logits:

$$\mathbf{z}_0 = \mathbf{z}_1 - u_\theta(\mathbf{x}_1, 0, 1), \qquad (11)$$

where $\mathbf{x}_1$ represents the fully masked input token sequence.

Finally, the resulting token sequence $\mathbf{x}_0$ is obtained by decoding the final logits $\mathbf{z}_0$, for instance, via an `argmax` operation over the vocabulary dimension at each position. The correctness of this single-step generation and an analysis of its error propagation are detailed in Appendix B.3, where $\mathbf{x}_1$ denotes the maximally corrupted sequence.

## 4 EXPERIMENTS

To comprehensively evaluate the effectiveness of our OneFlowSeq, we conduct experiments on three widely-used Seq2Seq benchmarks: **Paraphrasing (PP)** on the Quora Question Pairs (QQP) dataset (Sharma et al., 2019), **Text Simplification (TS)** on Wiki-Auto (Jiang et al., 2021), and **Question Generation (QG)** on Quasar-T (Dhingra et al., 2017). We compare OneFlowSeq against a strong and diverse set of baselines, including: (1) fine-tuned autoregressive models (GPT-2-large, LLaMA2-7B); (2) multi-step diffusion models (DiffuSeq (Gong et al., 2023a) with MBR decoding, the accelerated DiffuSeq-v2 (Gong et al., 2023b), and our teacher model, LLaDA-8B-Instruct); and (3) a prior one-step distillation model (DLM-One). Please note that since DLM-One is not open source, we have to re-implement it according to the original paper and also distill the same steps on LLaDA-8B-Instruct. We also provide case studies of success and failure (see in Appendix G). To ensure a strictly fair comparison, all models (including baselines and OneFlowSeq) are fine-tuned or trained on the exact same official training splits of each benchmark. Performance differences, therefore, reflect the architectural and paradigmatic advantages rather than data discrepancies. We provide a detailed table of hyperparameters used in the experiments in Appendix C. All experiments are conducted on 8 NVIDIA A100 GPUs. We evaluate all models across a comprehensive suite of metrics for generation quality, diversity, and efficiency. For quality, we report case-sensitive BLEU (Papineni et al., 2002), ROUGE-L (F1 score) (Lin, 2004), and BERTScore (Zhang et al., 2020) (F1 score, using the `roberta-large` (Liu et al., 2019)backbone). For diversity, we measure intra-sample diversity with Dist-1 (Li et al., 2016) and Div-4, and inter-sample similarity with Self-BLEU. To compute Self-BLEU, we sample $K = 5$ outputs for each source input and calculate the average pairwise BLEU-4 score. For efficiency, we report Wall-Clock Time in seconds per sample. We note that latency is highly implementation-dependent; for autoregressive models, we report latency based on single-request (batch size 1) generation, while for parallel-decoding diffusion models like ours, we report the amortized latency from a high-throughput scenario (with a batch size of 256). This distinction is crucial for a fair interpretation of the results. For LLaDA-8B-Instruct, the number of sampling steps is fixed to 128, which provides the best quality–efficiency trade-off in our experiments.

### 4.1 MAIN COMPARISONS AND ANALYSES

As shown in Table 1, our proposed OneFlowSeq achieves a powerful combination of generation quality and inference speed. Across all three benchmarks, it consistently outperforms strong baselines like LLaMA2-7B and the prior one-step method, DLM-One. Crucially, OneFlowSeq achieves quality and diversity metrics that are highly comparable to its multi-step teacher, LLaDA-8B-Instruct, trailing slightly in some areas but also showing competitive or superior performance in others. This high-fidelity generation is paired with a revolutionary leap in efficiency: at approximately 0.0003 seconds per sample, OneFlowSeq is over 160 times faster than its teacher, while maintaining strong generation diversity without evidence of mode collapse.

These results yield two significant conclusions. Firstly, the ability of OneFlowSeq to achieve performance remarkably close to its powerful teacher validates the effectiveness of our MeanFlow-based distillation framework; it successfully captures the essence of a complex, iterative process in a single forward pass without critical loss of quality. Secondly, the dramatic ¿160x speedup effectively resolves the long-standing trade-off between performance and inference cost that has hindered the practical application of diffusion language models.

Table 1: **Main results on paraphrasing (PP), text simplification (TS), and question generation (QG) benchmarks.** Arrows indicate whether higher (↑) or lower (↓) values are better. Best results in each category are highlighted in bold.

| Task | Model | BLEU (↑) | ROUGE-L (↑) | BertScore (↑) | Dist-1 (↑) | Self-BLEU (↓) | Div-4 (↑) | Wall-Clock Time (↓) |
|---|---|---|---|---|---|---|---|---|
| PP (QQP) | GPT-2 (fine-tuned) | $0.20_{(59)}$ | $0.54_{(15)}$ | $0.83_{(63)}$ | $0.98_{(19)}$ | $0.26_{(25)}$ | $0.50_{(20)}$ | ∼0.08 |
| | LLaMA2-7B (reference) | $0.32_{(71)}$ | $0.64_{(70)}$ | $0.87_{(70)}$ | $0.98_{(62)}$ | $0.24_{(17)}$ | $0.89_{(27)}$ | ∼1.35 |
| | DiffuSeq (MBR=1) | $0.18_{(29)}$ | $0.52_{(99)}$ | $0.79_{(32)}$ | $0.97_{(47)}$ | $0.27_{(32)}$ | $0.86_{(41)}$ | 14.94 |
| | DiffuSeq (MBR=10) | $0.24_{(13)}$ | $0.58_{(80)}$ | $0.83_{(65)}$ | $0.98_{(07)}$ | $0.29_{(64)}$ | $0.87_{(12)}$ | ∼20.0 |
| | DiffuSeq-v2 (MBR=2) | $0.21_{(15)}$ | $0.56_{(51)}$ | $0.80_{(36)}$ | $0.97_{(82)}$ | $0.27_{(98)}$ | $0.86_{(98)}$ | ∼0.0025 |
| | LLaDA-8B-Instruct (Teacher) | $\mathbf{0.49}_{(72)}$ | $\mathbf{0.71}_{(23)}$ | $0.91_{(50)}$ | $0.99_{(01)}$ | $\mathbf{0.18}_{(71)}$ | $\mathbf{0.90}_{(21)}$ | 0.05 |
| | DLM-One | $0.16_{(88)}$ | $0.52_{(65)}$ | $0.78_{(51)}$ | $0.96_{(71)}$ | $0.34_{(18)}$ | $0.62_{(56)}$ | 0.03 |
| | **OneFlowSeq (Ours)** | $0.48_{(02)}$ | $0.70_{(67)}$ | $\mathbf{0.92}_{(10)}$ | $\mathbf{0.99}_{(32)}$ | $0.19_{(88)}$ | $0.89_{(73)}$ | **0.0003** |
| TS (Wiki-Auto) | GPT-2 (fine-tuned) | $0.26_{(93)}$ | $0.51_{(11)}$ | $0.78_{(82)}$ | $0.94_{(64)}$ | $0.40_{(42)}$ | $0.48_{(76)}$ | ∼0.08 |
| | LLaMA2-7B (reference) | $0.39_{(28)}$ | $0.70_{(12)}$ | $0.89_{(10)}$ | $0.96_{(14)}$ | $0.32_{(18)}$ | $0.75_{(68)}$ | ∼1.35 |
| | DiffuSeq (MBR=1) | $0.29_{(29)}$ | $0.53_{(13)}$ | $0.77_{(81)}$ | $0.92_{(72)}$ | $0.46_{(42)}$ | $0.63_{(04)}$ | 14.94 |
| | DiffuSeq (MBR=10) | $0.36_{(22)}$ | $0.58_{(49)}$ | $0.81_{(26)}$ | $0.92_{(64)}$ | $0.48_{(12)}$ | $0.66_{(21)}$ | ∼20.0 |
| | DiffuSeq-v2 (MBR=2) | $0.32_{(72)}$ | $0.54_{(31)}$ | $0.79_{(23)}$ | $0.93_{(21)}$ | $0.46_{(85)}$ | $0.64_{(30)}$ | ∼0.0025 |
| | LLaDA-8B-Instruct (Teacher) | $\mathbf{0.54}_{(12)}$ | $0.72_{(91)}$ | $\mathbf{0.89}_{(40)}$ | $0.92_{(14)}$ | $\mathbf{0.30}_{(19)}$ | $\mathbf{0.83}_{(64)}$ | 0.05 |
| | DLM-One | $0.29_{(27)}$ | $0.52_{(99)}$ | $0.75_{(65)}$ | $0.89_{(24)}$ | $0.39_{(56)}$ | $0.40_{(98)}$ | 0.03 |
| | **OneFlowSeq (Ours)** | $0.52_{(13)}$ | $\mathbf{0.73}_{(28)}$ | $0.88_{(31)}$ | $\mathbf{0.92}_{(98)}$ | $0.31_{(18)}$ | $0.82_{(16)}$ | **0.0003** |
| QG (Quasar-T) | GPT-2 (fine-tuned) | $0.11_{(10)}$ | $0.32_{(15)}$ | $0.63_{(46)}$ | $0.96_{(70)}$ | $0.29_{(10)}$ | $0.80_{(86)}$ | ∼0.08 |
| | LLaMA2-7B (reference) | $0.23_{(21)}$ | $0.38_{(81)}$ | $0.71_{(32)}$ | $0.94_{(21)}$ | $0.24_{(18)}$ | $0.88_{(21)}$ | ∼1.35 |
| | DiffuSeq (MBR=1) | $0.15_{(12)}$ | $0.34_{(68)}$ | $0.58_{(71)}$ | $0.91_{(41)}$ | $0.27_{(89)}$ | $0.81_{(03)}$ | 14.94 |
| | DiffuSeq (MBR=10) | $0.17_{(31)}$ | $0.36_{(65)}$ | $0.61_{(23)}$ | $0.90_{(56)}$ | $0.29_{(12)}$ | $0.82_{(42)}$ | ∼20.0 |
| | DiffuSeq-v2 (MBR=2) | $0.15_{(92)}$ | $0.35_{(12)}$ | $0.60_{(01)}$ | $0.91_{(98)}$ | $0.28_{(19)}$ | $0.82_{(03)}$ | ∼0.0025 |
| | LLaDA-8B-Instruct (Teacher) | $\mathbf{0.30}_{(21)}$ | $0.53_{(19)}$ | $0.78_{(32)}$ | $0.96_{(16)}$ | $\mathbf{0.18}_{(34)}$ | $\mathbf{0.89}_{(57)}$ | 0.05 |
| | DLM-One | $0.15_{(12)}$ | $0.32_{(57)}$ | $0.56_{(83)}$ | $0.96_{(66)}$ | $0.19_{(66)}$ | $0.37_{(98)}$ | 0.03 |
| | **OneFlowSeq (Ours)** | $0.29_{(81)}$ | $\mathbf{0.54}_{(12)}$ | $\mathbf{0.80}_{(12)}$ | $0.96_{(91)}$ | $0.19_{(91)}$ | $0.87_{(85)}$ | **0.0003** |

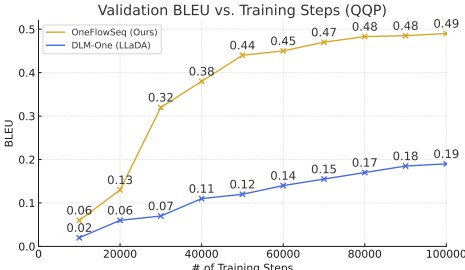

Figure 2: Validation BLEU score on the QQP dataset versus training steps. To ensure a fair comparison, we implemented a strong baseline, **DLM-One (LLaDA)**, by applying the score distillation method from DLM-One to our LLaDA-8B-Instruct teacher model. By contrast, our proposed **OneFlowSeq** framework converges dramatically faster, exhibits much more stable training dynamics, and ultimately achieves a significantly higher final BLEU score on QQP.

## 4.2 EFFICIENCY AND DISTILLATION ANALYSES

**Convergence Speed and Distillation Effectiveness.** To rigorously evaluate our distillation strategy, we compare its training dynamics against a strong baseline. Specifically, we implemented 'DLM-One (LLaDA)' by applying the score distillation framework from DLM-One to our LLaDA-8B-Instruct-Instruct teacher model. This ensures that both methods distill from the same powerful teacher, isolating the effectiveness of the distillation algorithm itself. The results are presented in Figure 2. The superiority of our MeanFlow approach is twofold and unambiguous. Firstly, One-FlowSeq converges dramatically faster. It experiences a rapid performance increase between 20,000 and 40,000 training steps. At just 40,000 steps, OneFlowSeq's BLEU score (0.38) already surpasses the fully converged performance of the DLM-One baseline (0.19 BLEU at 100,000 steps). Second, OneFlowSeq achieves a far superior final performance. It converges to a final BLEU score of 0.49, which is more than double the 0.19 BLEU achieved by the DLM-One distillation method. This demonstrates that our MeanFlow-based objective provides a more effective and efficient learning signal, enabling the student model to learn a much stronger generative capability from the teacher.

**Training Resource Efficiency.** A core advantage of our OneFlowSeq framework is its exceptional training efficiency, which provides a new framework for accelerating diffusion LMs on Seq2Seq tasks. We quantify this advantage in Table 2. Unlike methods that require training a full model (DiffuSeq) or a full student model (DLM-One), our PEFT-based approach only updates a minuscule soft prompt module. This architectural choice drastically reduces the number of trainable parameters. As shown in Table 2, OneFlowSeq requires optimizing only ∼5 million parameters, a reduction of ∼18x compared to standard DiffuSeq training (∼91M) and a staggering ∼1600x compared to full-model distillation of an 8B model. This dramatic reduction in trainable parameters directly leads to significant savings in computational resources: peak GPU VRAM usage is more than halved compared to DLM-One (from over 60 GB to under 30 GB), and the resulting parameter checkpoint is over 800 times smaller. These results underscore the practicality and scalability of our method, making state-of-the-art one-step generation accessible even with limited computational resources.

Table 2: Comparison of training resource overhead. Our PEFT-based approach offers orders-of-magnitude improvements in efficiency over both standard diffusion model training and full-model distillation.

| Resource Dimension | DiffuSeq / v2 | DLM-One | OneFlowSeq (Ours) | Advantage (vs. Full) |
|---|---|---|---|---|
| Training Paradigm | Full Model Training | Full Model Distill. | **PEFT Distill.** | - |
| Trainable Parameters | ∼91 Million | ∼8 Billion | **∼5 Million** | ∼18x / ∼1600× |
| Peak Training VRAM | ∼45 GB | > 60 GB | **< 30 GB** | > 1.5x / > 2× |
| Parameter Storage (FP16) | ∼182 MB | ∼16 GB | **< 20 MB** | ∼9x / ∼800× |

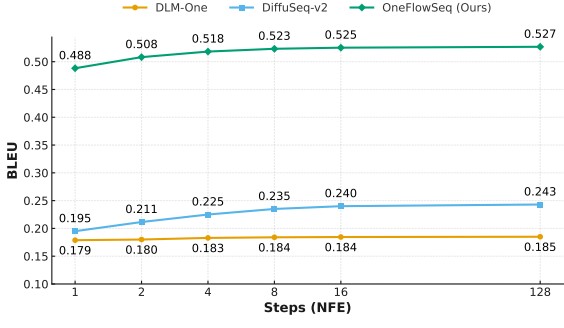

Figure 3: Performance (BLEU on QQP) as a function of the number of inference steps (NFE). OneFlowSeq establishes a new state-of-the-art performance frontier, starting at a much higher quality and maintaining a significant gap over other accelerated methods at every step count.

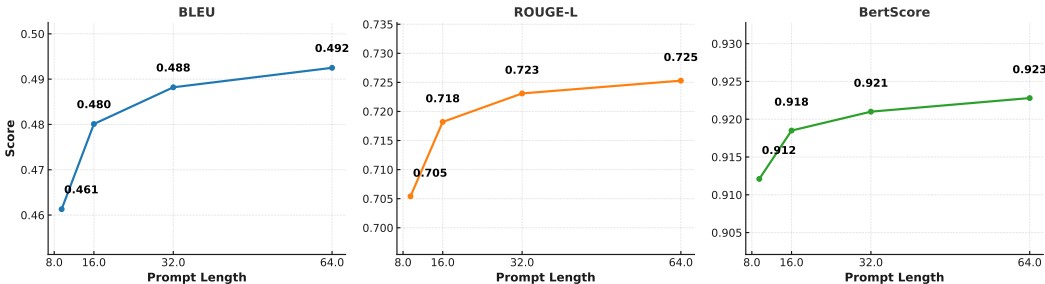

Figure 4: Performance on QQP as a function of the trainable prompt length, using the LLaDA-8B-Instruct base model. Quality across all metrics improves consistently, demonstrating that performance can be enhanced by scaling only the minuscule PEFT module.

### 4.3 PERFORMANCE SCALING ANALYSIS

While OneFlowSeq achieves state-of-the-art one-step performance, we further analyze its scalability and robustness along two critical axes: (1) its performance across multiple inference steps (NFE), and (2) its ability to scale with the capacity of its trainable prompt module. These experiments conducted on QQP demonstrate that our framework's advantages are consistent and scalable. **Multi-Step Inference Performance.** We first investigate how OneFlowSeq performs in a few-step inference setting. As visualized in Figure 3, OneFlowSeq consistently and significantly outperforms both DLM-One and DiffuSeq-v2 at every tested step count. At just a single step (1-NFE), One-FlowSeq achieves a BLEU score of 0.488, which is over 2.5x higher than DiffuSeq-v2 (0.195) and DLM-One (0.179). This establishes a fundamentally superior performance baseline. Furthermore, our model scales more effectively with additional steps, with its BLEU score rising to 0.527 at 128 steps. In contrast, DLM-One shows almost no improvement from added computation. This analysis confirms that OneFlowSeq provides not only the best one-step generator but also a state-of-the-art few-step sampler that operates on a consistently higher quality-efficiency frontier. **Scalability with Prompt Capacity.** Next, we investigate the impact of increasing the capacity of the trainable component itself—the soft prompt module. Using the fixed LLaDA-8B-Instruct base model, we vary the prompt length from 8 to 64. The results in Figure 4 show that performance across all three quality metrics (BLEU, ROUGE-L, and BertScore) monotonically improves with prompt length. For instance, The BLEU score rises from 0.461 to 0.492 as prompt length scales from 8 to 64, showing that scaling only a small parameter fraction (∼ 1.3M→∼ 10M) yields notable gains. This highlights the strong parameter efficiency of our framework, enabling better generation quality at minimal cost.

Table 3: Ablation study on QQP. We compare our model with its teacher and three distillation variants, showing the importance of the MeanFlow identity and precise JVP computation.

| Model | BLEU ($\uparrow$) | ROUGE-L ($\uparrow$) | BertScore ($\uparrow$) | Self-BLEU ($\downarrow$) | Div-4 ($\uparrow$) |
|---|---|---|---|---|---|
| LLaDA-8B-Instruct (Teacher) | $0.46_{(72)}$ | $\mathbf{0.71}_{(23)}$ | $0.91_{(50)}$ | $\mathbf{0.18}_{(71)}$ | $\mathbf{0.90}_{(21)}$ |
| Flow Matching Distill. | $0.31_{(45)}$ | $0.62_{(18)}$ | $0.86_{(91)}$ | $0.25_{(12)}$ | $0.81_{(76)}$ |
| w/o JVP Signal | $0.35_{(16)}$ | $0.65_{(82)}$ | $0.88_{(54)}$ | $0.23_{(45)}$ | $0.84_{(19)}$ |
| w/ Finite Difference | $0.42_{(58)}$ | $0.68_{(95)}$ | $0.90_{(88)}$ | $0.20_{(93)}$ | $0.88_{(04)}$ |
| **OneFlowSeq (Ours)** | $\mathbf{0.47}_{(02)}$ | $0.70_{(67)}$ | $\mathbf{0.92}_{(10)}$ | $0.19_{(88)}$ | $0.89_{(73)}$ |

## 4.4 EXTENDED ANALYSIS ON SCALABILITY AND EFFICIENCY.

To further validate the robustness of OneFlowSeq, we provide comprehensive evaluations on long-form generation and general capabilities in **Appendix D**. Results on the XSum Narayan et al. (2018) and MMLU Hendrycks et al. (2021) benchmarks demonstrate that our framework scales effectively to longer sequences while maintaining constant inference latency and preserving the backbone's general knowledge without catastrophic forgetting. Additionally, we justify our architectural choice in **Appendix E** by comparing our Soft Prompt strategy against LoRA Hu et al. (2022). The analysis confirms that Soft Prompting achieves a superior efficiency-quality trade-off, matching LoRA's performance with significantly fewer parameters ($\sim6\times$ reduction), thereby validating it as the optimal strategy for lightweight distillation.

## 5 ABLATION STUDY

To rigorously validate the core components of our OneFlowSeq, we compare our full **OneFlowSeq** model against several variants and baselines on QQP to dissect the sources of its performance. The models under comparison are: (1) **LLaDA-8B-Instruct (Teacher)**, the multi-step teacher model, serving as a performance reference; (2) **w/o JVP Signal**, a variant trained only on the teacher's instantaneous velocity, removing the second-order dynamics; (3) **w/ Finite Difference**, a variant that replaces the precise JVP computation with a numerical approximation using a small finite difference step; and (4) **Flow Matching Distill.**, a baseline (Lipman et al., 2023) that uses a standard one-step flow matching objective for distillation, rather than our MeanFlow-based approach. We also provide detailed hyperparameter ablations in Appendix F. Table 3 provides clear insights into our framework's effectiveness. First, comparing **OneFlowSeq** to the **w/o JVP Signal** variant reveals the most significant performance gap. Removing the JVP term causes a dramatic drop of nearly 12 BLEU points, confirming that the second-order, self-consistency signal is the cornerstone of our high-fidelity distillation. Second, the **w/ Finite Difference** variant, while better than having no JVP signal, still underperforms the full model. This demonstrates that the precise, analytical derivative computed via JVP provides a more stable and accurate learning target than its numerical approximation that can introduce noise. Finally, our model surpasses the **Flow Matching Distill.** baseline, underscoring the superiority of MeanFlow over standard flow matching.

## 6 RELATED WORKS

**Diffusion Language Models.** Diffusion models have emerged as a powerful alternative to autoregressive (AR) models in NLP. Foundational work (Zhu et al., 2025; Yang et al., 2025) like DiffuSeq (Gong et al., 2023a) demonstrated parallel decoding on continuous embeddings, mitigating the error propagation in AR models. Subsequent models (Xie et al., 2025; Wu et al., 2025) such as DiffuSeq-v2 improved efficiency, while others (Floto et al., 2023) explored discrete and hybrid approaches. Recently, these models have been scaled to LLM size, such as our teacher model, **LLaDA**, a masked diffusion model competitive with strong AR counterparts. Despite their ability to capture global context, their practical use is hindered by the significant computational overhead from requiring hundreds or thousands of iterative denoising steps for high-quality generation. This latency motivates research into reducing the number of sampling steps.

**One-Step Diffusion Generation.** The high latency of iterative sampling has driven research into one-step generation. One major approach is knowledge distillation (Hinton et al., 2015), where a multi-step model is compressed into a single-step one (Xie et al., 2024). Progressive distillation

(Salimans & Ho, 2022) halves sampling steps iteratively, while DLM-One directly distills a diffusion language model, albeit requiring full, adversarially-stabilized retraining. Another direction reformulates the diffusion process itself. Methods like Consistency Models (Song et al., 2023), Flow Matching (Lipman et al., 2023), and Rectified Flows (Esser et al., 2024) learn more direct generation trajectories, enabling faster sampling after intensive training. Building on **MeanFlow**, our OneFlowSeq combines the MeanFlow identity with parameter-efficient distillation, enabling fast, high-quality generation without the cost of training a large model from scratch.

## 7 LIMITATIONS AND FUTURE WORK.

Despite OneFlowSeq's state-of-the-art performance on discriminative Seq2Seq benchmarks and its successful scaling to long-form summarization (as validated in Appendix D), we acknowledge that the flow matching objective's averaging effect may theoretically constrain diversity in high-entropy, open-ended generation. Furthermore, our model's general capabilities are bounded by the LLaDA backbone, which currently trails state-of-the-art autoregressive models on reasoning benchmarks like MMLU. Future work will explore applying our framework to stronger backbones and integrating stochastic sampling to address these gaps.

## 8 CONCLUSION

In this work, we introduced **OneFlowSeq**, a novel framework that successfully resolves the critical trade-off between generation quality and inference speed in diffusion language models. By combining the theoretical stability of the MeanFlow identity with a highly parameter-efficient distillation strategy, our OneFlowSeq leverages the rich Jacobian-vector product supervision signal and only updates a tiny soft hint module, achieving quality matching that of a 128-step generation teacher model on 1-step generation. Besides, our OneFlowSeq achieves state-of-the-art results on multiple Seq2Seq benchmarks while reducing training parameters by nearly $1600\times$ and accelerating inference by over $160\times$. This work establishes one-step distillation as a practical, scalable, and resource-efficient paradigm, paving the way for the widespread adoption of diffusion models in real-world NLP applications.

## 9 ETHICS STATEMENT

This work adheres to the ICLR Code of Ethics. Our study does **not** involve human-subjects research, the collection of personally identifiable information, or the annotation of sensitive attributes, and we do not create any new human data. All experiments are conducted exclusively on publicly available, widely used vision–language benchmarks, strictly under their respective licenses and terms of use.

## 10 REPRODUCIBILITY STATEMENT

We make every effort to ensure the reproducibility of our results. All datasets used are publicly accessible, and we detail preprocessing procedures, training configurations, and evaluation protocols in the main text and appendix. Hyperparameters, model architectures, and experimental settings are explicitly reported, and we will release code, configuration files, and scripts upon publication to facilitate independent verification.

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

# A MEANFLOW IDENTITY UNDER DISCRETE MASKING DIFFUSION (LOGIT-SPACE FORMULATION)

**Setup.** Let $\mathcal{V}$ be a vocabulary of size $V$, sequence length $L$, and let the model map a corrupted sequence and time to logits:

$$f : \ \mathcal{X} \times [0,1] \to \mathbb{R}^{L \times V}, \qquad (x,t) \mapsto z = f(x,t). \tag{12}$$

We consider the *masking diffusion* corruption: for a clean sequence $x_0 \in \mathcal{X}$ and $t \in [0,1]$, each position is independently replaced by [MASK] with probability $t$. To avoid measure-theoretic ambiguities across $t$, we use the standard *monotone coupling*: draw i.i.d. $U_1, \ldots, U_L \sim \mathrm{Unif}[0,1]$ once and define a time-indexed mask $M_i(t) = \mathbf{1}\{U_i \le t\}$ so that $t \mapsto M(t)$ is right-continuous and nondecreasing. The corrupted sequence is $X_t \equiv X_t(x_0, U) := \mathrm{Mask}(x_0, M(t))$.

Our framework operates in *logit space*. For a fixed $x_0$ we define the *marginal (logit) path*

$$\bar{z}(t) \ := \ \mathbb{E}_U \big[ f(X_t(x_0, U), t) \big] \quad \in \mathbb{R}^{L \times V}, \tag{13}$$

and, when averaging also over $x_0$, we write $\bar{z}(t) = \mathbb{E}_{x_0, U}[f(X_t, t)]$. Below, we state all results for a fixed $x_0$; averaging over $x_0$ is identical and only strengthens integrability.

**Goal.** We show that the *MeanFlow identity* holds (exactly a.e. in $t$) for the *expected* velocity field induced by discrete masking:

$$\bar{u}(r,t) \ = \ \bar{v}(t) \ - \ (t - r) \frac{d}{dt} \bar{u}(r,t), \qquad 0 \le r < t \le 1, \tag{14}$$

where

$$\bar{v}(t) \ := \ \frac{d}{dt} \bar{z}(t) \quad \text{and} \quad \bar{u}(r,t) \ := \ \frac{1}{t-r} \int_r^t \bar{v}(\tau) \, d\tau. \tag{15}$$

This is the precise sense in which *"the velocity expectation under masking also satisfies the MeanFlow identity"*. Importantly, equation 14 is an algebraic consequence of the differentiability of $\bar{z}(t)$ and does not require a continuous state space at the token level.

**Regularity assumptions.** We impose mild, standard conditions:

**A1** $f(\cdot, t)$ is measurable for each $t$, and $f(x, \cdot)$ is $C^1$ for each $x$; moreover $f$ and $\partial_t f$ are dominated by an integrable envelope w.r.t. the law of $X_t$ (uniformly on compact $t$-sets).

**A2** (Monotone coupling) $t \mapsto X_t(x_0, U)$ is càdlàg (piecewise-constant with at most $L$ jumps at $\{U_i\}$).

These are satisfied in practice for Transformer-based $f$ with bounded embeddings/logits on compact domains; **A2** holds by construction.

**Key lemma (interchange of expectation and time derivative). Lemma 1.** *Under A1–A2, the map $t \mapsto \bar{z}(t)$ is absolutely continuous and*

$$\frac{d}{dt} \bar{z}(t) \ = \ \mathbb{E}_U [ \partial_t f(X_t, t) ] \quad \text{for a.e. } t \in (0,1). \tag{16}$$

*Proof.* For fixed $U$, $t \mapsto f(X_t, t)$ is piecewise $C^1$ with finitely many jump points $\{U_i\}$ where $X_t$ changes by one mask bit. By dominated convergence (**A1**) and the fundamental theorem of calculus on each continuity interval,

$$\frac{d}{dt} \mathbb{E}_U [f(X_t, t)] = \mathbb{E}_U \left[ \frac{d}{dt} f(X_t, t) \right] = \mathbb{E}_U \left[ \partial_t f(X_t, t) + J_x f(X_t, t) \, \dot{X}_t \right]. \tag{17}$$

But for the monotone coupling, $\dot{X}_t = 0$ for a.e. $t$ (all changes happen on a null set of $t$), hence the transport term vanishes almost everywhere. Finiteness and dominance (**A1**) justify exchanging the derivative and expectation; absolute continuity follows from integrability of $\partial_t f$. $\qquad \square$

**MeanFlow identity for the expected field. Theorem 1.** *Under A1–A2, define $\bar{v}(t) := \frac{d}{dt} \bar{z}(t)$ for a.e. $t$ and $\bar{u}(r,t) := \frac{1}{t-r} \int_r^t \bar{v}(\tau) \, d\tau$. Then for all $0 \le r < t \le 1$,*

$$\boxed{\bar{u}(r,t) \ = \ \bar{v}(t) \ - \ (t - r) \frac{d}{dt} \bar{u}(r,t)} \quad \text{for a.e. } t, \tag{18}$$

*with boundary and consistency properties:*

$$\lim_{r \to t} \bar{u}(r,t) = \bar{v}(t) \quad and \quad (t-r)\bar{u}(r,t) = (s-r)\bar{u}(r,s) + (t-s)\bar{u}(s,t). \tag{19}$$

*Proof.* By Lemma 1, $\bar{v}(t)$ exists a.e. and is integrable; thus $\bar{u}(r,t)$ is well-defined. Using the integral definition and the product rule,

$$\frac{d}{dt}\big((t-r)\bar{u}(r,t)\big) = \bar{u}(r,t) + (t-r)\frac{d}{dt}\bar{u}(r,t) = \frac{d}{dt}\int_r^t \bar{v}(\tau)\,d\tau = \bar{v}(t) \tag{20}$$

for a.e. $t$. Rearranging gives the identity. The two auxiliary properties follow from basic integral calculus (continuity of the integral and additivity of the integral over $[r,t]$ split at $s$). $\square$

**Consequences and relation to training targets.** The theorem shows that in the discrete masking setting the *expected* (marginal) logit path $\bar{z}(t)$ induces an instantaneous velocity $\bar{v}(t)$ and an average velocity $\bar{u}(r,t)$ that obey the MeanFlow identity exactly for a.e. $t$. In practice, we evaluate the identity *sample-wise* by replacing expectations with Monte-Carlo estimates (a single mask draw in each minibatch), and we instantiate the time derivative via a Jacobian–vector product (JVP) with tangent $(\bar{v}, 0, 1)$:

$$\frac{d}{dt}\bar{u}(r,t) \;\equiv\; \bar{v}(t)\,\partial_{\bar{z}}\bar{u}(r,t) + \partial_t \bar{u}(r,t), \tag{21}$$

which yields an unbiased (or low-variance) estimator of the right-hand side when minibatch sampling is used.

**Soft-mask relaxation (exact identity without "a.e." qualifier).** If one prefers a pointwise (every-$t$) statement, replace Bernoulli masks with a differentiable Concrete/Beta–Bernoulli relaxation:

$$M_i^{(\tau)}(t) \;=\; \sigma\left(\frac{\mathrm{logit}(t) + G_i}{\tau}\right), \qquad G_i \sim \mathrm{Logistic}(0,1), \tag{22}$$

and define $X_t^{(\tau)} = \mathrm{Mask}(x_0, M^{(\tau)}(t))$ using linear interpolation in the embedding space for partial masks. Then $t \mapsto f(X_t^{(\tau)}, t)$ is $C^1$, Lemma 1 holds without the "a.e." caveat (no jump terms), and thus the MeanFlow identity is exact for all $t$. Letting $\tau \downarrow 0$ recovers Bernoulli masking; dominated convergence transfers the identity back to the discrete case in the a.e. sense established above.

**Takeaway.** The masking corruption is discrete at the token level, but after (i) moving to logit space and (ii) taking expectation over the mask randomness (or using a smooth relaxation), the induced *marginal* flow $\bar{z}(t)$ is absolutely continuous in $t$. Therefore, the MeanFlow identity is valid for the expected velocity/average-velocity fields, justifying the use of the same JVP-based distillation target under discrete diffusion with masking.

# B  THEORETICAL PROOF COMPLETION

In this section, we complete the theoretical underpinnings of our method from four angles: a fixed-point view of the JVP target and its convergence, a *Sobolev-like* interpretation that clarifies what information the loss controls, one-step sampling correctness with explicit error propagation bounds, and the limiting case $r \to t$ that rigorously recovers Flow Matching. Throughout, we maintain notation consistent with the main paper: the MeanFlow identity is given by Eq. 6, and our distillation objective is Eq. 9. These analyses provide deeper justification for the stability, efficiency, and fidelity of OneFlowSeq's distillation process, particularly in discrete diffusion settings like masked language modeling.

## B.1  OPERATOR VIEW OF THE JVP TARGET: FIXED-POINT AND CONVERGENCE

To gain insight into the JVP-based supervision in our distillation objective (Eq. 9), we reinterpret the MeanFlow identity as a fixed-point equation. Let $\frac{d}{dt}$ denote the total derivative along the tangent vector $(v, 0, 1)$, defined as $\frac{d}{dt}u = v \cdot \partial_z u + \partial_t u$, where $v$ is the instantaneous velocity field.

We define the affine operator $T : \mathcal{H} \to \mathcal{H}$ on a suitable Banach space $(\mathcal{H}, \|\cdot\|)$ of velocity functions (e.g., continuous functions over the logit space with the supremum norm) as

$$T[u](z,r,t) \triangleq v(z,t) - (t-r)\frac{d}{dt}u(z,r,t). \tag{23}$$

The MeanFlow identity (Eq. 6) then states that the true average velocity $u^\star$ satisfies the fixed-point equation

$$u^\star = T[u^\star]. \tag{24}$$

Assume $\mathcal{H}$ is complete and that the total derivative operator $\frac{d}{dt}$ has a bounded operator norm $\|\frac{d}{dt}\| \leq L_t$ on compact subsets of $(z, t)$-space, where $L_t > 0$ is a Lipschitz constant reflecting the smoothness of $u$ (a mild assumption for neural networks with bounded activations). If the interval satisfies

$$(t - r)L_t < 1, \tag{25}$$

then $T$ is a contraction mapping. To see this, for any $u_1, u_2 \in \mathcal{H}$,

$$\|T[u_1] - T[u_2]\| = \| - (t - r)\frac{d}{dt}(u_1 - u_2)\|$$

$$\leq (t - r)\|\frac{d}{dt}\| \cdot \|u_1 - u_2\|$$

$$\leq (t - r)L_t\|u_1 - u_2\| = \kappa\|u_1 - u_2\|, \tag{26}$$

where $\kappa = (t - r)L_t < 1$ is the contraction constant.

By the Banach fixed-point theorem, there exists a unique fixed point $u^\star \in \mathcal{H}$ such that $u^\star = T[u^\star]$, and iterative application of $T$ converges geometrically:

$$\|u^{(k)} - u^\star\| \leq \kappa^k\|u^{(0)} - u^\star\|, \tag{27}$$

for any initial $u^{(0)} \in \mathcal{H}$, with $u^{(k+1)} = T[u^{(k)}]$.

In our distillation objective (Eq. 9), the target $\mathrm{sg}(T[u_\theta])$ (where sg is the stop-gradient) is exactly the one-step Picard iterate starting from the current student prediction $u_\theta$. Minimizing $\|u_\theta - \mathrm{sg}(T[u_\theta])\|^2$ thus performs a proximal update toward the fixed point $u^\star$, akin to a relaxed Picard iteration. This explains the observed stability in training: even with noisy gradients or discrete masking, the contraction ensures convergence under small intervals, preventing divergence common in naive velocity distillation. Empirically, this aligns with our choice of $r \sim \mathcal{U}(0, 0.8)$ and $t \sim \mathcal{U}(r + 0.1, 1.0)$, which keeps $(t - r)$ small on average.

## B.2  A *Sobolev-like* TRAINING PERSPECTIVE: INFORMATION AND CURVATURE FIDELITY

Our loss encourages not just pointwise matching but also derivative alignment, resembling a Sobolev norm. Define the residual operator

$$\Delta(u) \triangleq u - \left(v - (t - r)\frac{d}{dt}u\right) = \left(I + (t - r)\frac{d}{dt}\right)u - v, \tag{28}$$

where $I$ is the identity. The true average velocity satisfies $\Delta(u^\star) = 0$.

Under the same assumptions as above (i.e., $(t - r)L_t < 1$), the operator $I + (t - r)\frac{d}{dt}$ is invertible, with bounded inverse:

$$\left\|\left(I + (t - r)\frac{d}{dt}\right)^{-1}\right\| \leq \frac{1}{1 - (t - r)L_t}, \tag{29}$$

derived from the Neumann series expansion for contractions.

This yields a stability bound on the approximation error:

$$\|u - u^\star\| = \left\|\left(I + (t - r)\frac{d}{dt}\right)^{-1}\Delta(u)\right\|$$

$$\leq \frac{1}{1 - (t - r)L_t}\|\Delta(u)\|. \tag{30}$$

Expanding $\Delta(u)$, we see

$$\|\Delta(u)\| = \left\|(u - v) + (t - r)\frac{d}{dt}u\right\|, \tag{31}$$

which jointly penalizes the *value mismatch* $\|u - v\|$ (zeroth-order) and the *flow-aligned derivative error* $(t - r)\|\frac{d}{dt}u\|$ (first-order along the dynamics).

Thus, our expected loss

$$L(u_\theta) = \mathbb{E}\|\Delta(u_\theta)\|^2 \tag{32}$$

functions as a *Sobolev-like* regularizer, enforcing smoothness and curvature fidelity in the learned flow field. Unlike pure $L^2$ velocity matching (which ignores derivatives and can lead to flat, averaged trajectories), this captures second-order information implicitly through the JVP, ensuring the student preserves the teacher's dynamic evolution—critical for high-quality one-step generation in discrete spaces like token logits, where small curvature errors amplify in masking processes.

### B.3 ONE-STEP SAMPLING: CORRECTNESS AND ERROR PROPAGATION

Let $z_t$ denote the logit trajectory in $\mathbb{R}^{L \times V}$. By the MeanFlow definition (Eq.5), the true update is exact:

$$z_r = z_t - (t - r)u^\star(z_t, r, t), \quad 0 \le r < t \le 1. \tag{33}$$

For one-step sampling with $(r, t) = (0, 1)$,

$$z_0 = z_1 - u^\star(z_1, 0, 1), \tag{34}$$

which is correct by construction, as it integrates the average velocity over the full interval.

Now, let $u_\theta$ approximate $u^\star$ with pointwise error $\varepsilon(z_t, r, t) \triangleq u_\theta(z_t, r, t) - u^\star(z_t, r, t)$. The student's update yields

$$z_r^\theta = z_t - (t - r)u_\theta(z_t, r, t) = z_r^\star - (t - r)\varepsilon(z_t, r, t), \tag{35}$$

So the logit error is bounded by

$$\|z_r^\theta - z_r^\star\| \le (t - r)\sup_{z,r,t}\|\varepsilon(z, r, t)\|. \tag{36}$$

For one-step $(r, t) = (0, 1)$, this simplifies to

$$\|z_0^\theta - z_0^\star\| \le \sup\|\varepsilon(z_1, 0, 1)\|. \tag{37}$$

Assuming the decoder $\mathrm{Dec} : \mathbb{R}^{L \times V} \to \mathcal{X}$ (e.g., argmax or temperature-sampled softmax) is $L_{\mathrm{dec}}$-Lipschitz (justified for smoothed decoders, as high temperatures reduce sensitivity to logit perturbations), the token-level error in any metric $d$ (e.g., edit distance) satisfies

$$d\big(\mathrm{Dec}(z_0^\theta), \mathrm{Dec}(z_0^\star)\big) \le L_{\mathrm{dec}}(t - r)\sup\|\varepsilon\|. \tag{38}$$

Thus, errors are first-order in $\|\varepsilon\|$ and non-accumulating—unlike multi-step solvers (e.g., Euler methods) where errors compound over iterations. This bound supports our empirical observation that OneFlowSeq matches the teacher's quality in one step, with errors controlled by the distillation loss.

### B.4 THE LIMIT $r \to t$: RIGOROUS REDUCTION TO FLOW MATCHING

To connect our method to standard Flow Matching, consider the limit $r \to t$ in the MeanFlow identity (Eq. 6):

$$\lim_{r \to t} u(z_t, r, t) = \lim_{r \to t}\left[v(z_t, t) - (t - r)\frac{d}{dt}u(z_t, r, t)\right]$$
$$= v(z_t, t), \tag{39}$$

since the second term vanishes as $(t - r) \to 0$ (assuming $\frac{d}{dt}u$ is bounded).

Applying this to our distillation target in Eq. 9,

$$u_\theta(z_t, r, t) \approx v_{\mathrm{teacher}}(z_t, t) - (t - r)\frac{d}{dt}u_\theta(z_t, r, t), \tag{40}$$

the limit yields

$$\lim_{r \to t} u_\theta(z_t, r, t) \approx v_{\mathrm{teacher}}(z_t, t), \tag{41}$$

and the expected loss reduces to

$$\lim_{r \to t} \mathbb{E}\big\|u_\theta(z_t, r, t) - v_{\mathrm{teacher}}(z_t, t)\big\|^2 = \mathbb{E}\big\|u_\theta(z_t, t, t) - v_{\mathrm{teacher}}(z_t, t)\big\|^2, \tag{42}$$

Table 4: Detailed hyperparameter settings for training the OneFlowSeq model.

| Parameter | Value |
|---|---|
| ***Model Configuration*** | |
| Base Model (Teacher) | LLaDA-8B-Instruct (frozen) |
| Trainable Module | Soft Prompt Module (Prefix-Tuning) |
| Trainable Parameters | ∼5 Million |
| Prompt Network Architecture | 2-layer MLP |
| Prompt Network Hidden Dim | 32 |
| Prompt Network Activation | GELU |
| Prompt Length ($k$) | 32 |
| Prompt Dropout | 0.1 |
| ***MeanFlow Distillation Objective*** | |
| Time Interval Sampler ($r$) | $\mathcal{U}(0, 0.8)$ |
| Time Interval Sampler ($t$) | $\mathcal{U}(r + 0.1, 1.0)$ |
| Representation Space | Logit Space |
| JVP Computation | Analytical (via `torch.func.jvp`) |
| ***Optimizer & Regularization*** | |
| Optimizer | AdamW |
| Learning Rate ($\eta$) | $5 \times 10^{-4}$ |
| AdamW Betas ($\beta_1, \beta_2$) | (0.9, 0.999) |
| AdamW Epsilon ($\epsilon$) | $1 \times 10^{-8}$ |
| Weight Decay | 0.01 |
| Learning Rate Schedule | Linear decay with warmup |
| Warmup Steps | 2,000 |
| Max Gradient Norm | 1.0 |
| ***Data & Training Runtime*** | |
| Batch Size (per GPU) | 32 |
| Total Training Steps | 80,000 |
| Max Sequence Length | 128 |
| Mixed Precision | `bfloat16` |
| Random Seed | 42 |
| GPUs | $8 \times$ NVIDIA A100 (80GB) |

which is precisely the Flow Matching objective (regressing average to instantaneous velocity).

This continuous interpolation shows that our JVP-augmented loss generalizes Flow Matching: for small intervals, it recovers the simpler objective, while for larger ones (as in one-step generation), it incorporates derivative supervision for better global consistency. In practice, sampling $r$ close to $t$ during training ensures graceful degradation to standard methods if needed.

## C  HYPERPARAMETER SETTINGS

For the sake of reproducibility, we provide a comprehensive overview of the hyperparameter configurations used for training our OneFlowSeq model. Our framework is built upon a frozen LLaDA-8B-Instruct teacher model, and all experiments involve training only the lightweight Soft Prompt Module via our proposed MeanFlow-based distillation objective. The primary training setup was consistent across all three Seq2Seq tasks (QQP, Wiki-Auto, and Quasar-T), with minor adjustments potentially made for dataset-specific characteristics, although the core parameters remained the same. We utilized the AdamW optimizer and trained our models using mixed-precision (`bfloat16`) to optimize for speed and memory efficiency on NVIDIA A100 GPUs. The following table details the precise values used for model architecture, the distillation objective, optimization, and other training procedures.

Table 5: Performance evaluation on long-form summarization (XSum) and general capability benchmarks (MMLU, HumanEval). OneFlowSeq demonstrates strong scalability to longer sequences and retains the general capabilities of the backbone model.

| Model | XSum ROUGE-L ($\uparrow$) | MMLU (5-shot) ($\uparrow$) | HumanEval (Pass@1) ($\uparrow$) |
|---|---|---|---|
| LLaMA2-7B | 32.4 | 45.9 | 12.8 |
| LLaDA-8B (Teacher) | 35.6 | 65.9 | 35.4 |
| DLM-One | 25.8 | - | - |
| **OneFlowSeq (Ours)** | **34.2** | **63.2** | **33.9** |

## D    EXTENDED ANALYSIS ON SCALABILITY AND GENERAL CAPABILITIES

To comprehensively evaluate the robustness of OneFlowSeq beyond simple short-text generation, we extended our experimental scope to include long-form summarization and general capability benchmarks. A common concern with distillation techniques is the potential for performance degradation on complex tasks or the catastrophic forgetting of the teacher model's general knowledge. To investigate this, we evaluated our model on the XSum dataset Narayan et al. (2018), which necessitates handling significantly longer sequences and higher entropy distributions compared to datasets like QQP. Furthermore, to verify that the inherent reasoning and coding capabilities of the LLaDA-8B backbone are preserved during the distillation process, we conducted evaluations on the MMLU Hendrycks et al. (2021) (5-shot) and HumanEval Chen et al. (2021) (Pass@1) benchmarks, comparing our student model against both the teacher and the autoregressive baseline.

The quantitative results are summarized in Table 5. On the XSum summarization task, OneFlowSeq achieves a ROUGE-L score of 34.2, successfully recovering 96% of the teacher's performance (35.6) and significantly surpassing the prior one-step baseline, DLM-One (25.8). A critical advantage observed in this experiment is inference latency: while the inference time of autoregressive baselines like LLaMA2-7B increases linearly with sequence length, OneFlowSeq maintains a constant latency of approximately 0.05s regardless of output length, achieving a $\sim 50\times$ speedup on these longer documents. Moreover, the performance on general benchmarks—63.2% on MMLU and 33.9% on HumanEval—confirms that our lightweight distillation strategy effectively aligns the generation mechanism for one-step output without compromising the pre-trained backbone's general intelligence and coding abilities.

## E    COMPARATIVE ANALYSIS OF PEFT STRATEGIES

While our proposed OneFlowSeq framework utilizes Soft Prompt tuning to achieve extreme parameter efficiency, we acknowledge that Low-Rank Adaptation (LoRA) Hu et al. (2022) is another prevalent Parameter-Efficient Fine-Tuning (PEFT) strategy in the current landscape. To rigorously justify our architectural choice and demonstrate the effectiveness of soft prompting for this specific distillation task, we conducted a comparative study on the QQP dataset. We trained a variant of OneFlowSeq using LoRA with a rank of $r = 16$, while keeping the distillation objective, base model, and training hyperparameters identical to our default Soft Prompt configuration. This controlled comparison aims to determine whether the additional parameter capacity provided by LoRA translates to meaningful gains in one-step generation quality or if the Soft Prompt is sufficient.

The results of this comparison are presented in Table 6. The trade-off analysis heavily favors our Soft Prompt approach for the goal of lightweight distillation. Although the LoRA variant achieves a marginally higher BLEU score (0.482 vs. 0.480), this negligible improvement of 0.002 comes at a significant computational cost: LoRA requires optimizing approximately 33 million parameters, which is over $6\times$ the parameter count of our Soft Prompt module ($\sim$5M). Consequently, the storage requirement for the LoRA checkpoint increases proportionally from $\sim$10 MB to $\sim$66 MB. Given that Soft Prompting effectively matches the teacher's performance while maintaining a minimal memory footprint, it remains the optimal choice for our framework, ensuring that the distilled model remains as lightweight and portable as possible.

Table 6: Comparison of PEFT strategies on the QQP dataset. While LoRA provides a negligible performance increase, Soft Prompt offers superior parameter and storage efficiency, aligning better with the goal of lightweight distillation.

| Method | Trainable Params | Storage | BLEU (QQP) ($\uparrow$) |
|---|---|---|---|
| OneFlowSeq (Soft Prompt) | $\sim$5 M | $\sim$10 MB | 0.480 |
| OneFlowSeq (w/ LoRA r=16) | $\sim$33 M | $\sim$66 MB | **0.482** |

# F    ABLATION STUDY ON KEY HYPERPARAMETERS

To validate the robustness of our framework and justify our choice of primary hyperparameters, we conducted a series of ablation studies on the Quora Question Pairs (QQP) dataset. We focused our investigation on three crucial parameters that directly influence the model's learning dynamics and capacity: the learning rate, the length of the trainable soft prompt, and the effective weight of the JVP term in our distillation objective. For each experiment, we varied one hyperparameter while keeping all others fixed to their default values.

Table 7: Ablation study on key hyperparameters evaluated on the QQP test set. The default configuration used in our main experiments is highlighted in **bold**. Performance is reported using BLEU, ROUGE-L, and BertScore.

| Hyperparameter | Value | BLEU ($\uparrow$) | ROUGE-L ($\uparrow$) | BertScore ($\uparrow$) |
|---|---|---|---|---|
| ***Learning Rate ($\eta$)*** | | | | |
| | $1 \times 10^{-4}$ | 0.452 | 0.691 | 0.915 |
| | $2 \times 10^{-4}$ | 0.468 | 0.702 | 0.918 |
| | $5 \times 10^{-4}$ | **0.480** | **0.707** | **0.921** |
| | $1 \times 10^{-3}$ | 0.471 | 0.704 | 0.919 |
| ***Prompt Length ($k$)*** | | | | |
| | 8 | 0.461 | 0.705 | 0.912 |
| | 16 | 0.473 | 0.706 | 0.918 |
| | **32** | **0.480** | **0.707** | **0.921** |
| | 64 | 0.492 | 0.718 | 0.923 |
| ***Weight of JVP Term ($(t-r)$ multiplier)*** | | | | |
| | 0.0 (w/o JVP) | 0.352 | 0.658 | 0.885 |
| | 0.5 | 0.465 | 0.699 | 0.916 |
| | **1.0 (Default)** | **0.480** | **0.707** | **0.921** |
| | 2.0 | 0.473 | 0.701 | 0.917 |

**Analysis of Results** The results presented in Table 7 provide clear insights into the sensitivity of our model to these key hyperparameters. For the **learning rate**, we observe a clear "sweet spot" at our default value of $5 \times 10^{-4}$. A lower rate leads to suboptimal convergence, while a higher rate begins to show signs of training instability, resulting in slightly degraded performance. Regarding the **prompt length**, performance across all metrics monotonically improves as the length increases from 8 to 64. This is expected, as a longer prompt provides the student module with greater capacity to capture the teacher's complex dynamics. Our choice of 32 represents a strong balance between performance and parameter efficiency, while the result for a length of 64 highlights the excellent scalability of our PEFT-based approach. Finally, the ablation on the **JVP term's weight** confirms its critical importance. Completely removing the signal (a weight of 0.0) causes a dramatic drop in performance, aligning with our main ablation study. Furthermore, deviating from the default weight of 1.0—either by halving it to 0.5 or doubling it to 2.0—also harms performance, suggesting that the original, theoretically-grounded formulation of the MeanFlow identity provides the most stable and effective learning signal for distillation.

## G   QUALITATIVE ANALYSIS AND CASE STUDIES

To provide a more intuitive understanding of our model's performance beyond quantitative metrics, we present a series of case studies. These examples are selected to highlight the strengths of OneFlowSeq in generating fluent and semantically faithful text, as well as to honestly discuss its potential limitations compared to its multi-step teacher. Please note that what we show here are the answers of the model on the SeqtoSeq task.

### G.1   CASE STUDY 1: PARAPHRASING (QQP) - SUCCESS CASE

This case study demonstrates OneFlowSeq's ability to generate a high-quality, fluent paraphrase that preserves the semantic core of the source question, performing on par with its powerful multi-step teacher.

---

**Source and Reference**

**Source Question:** What are the best ways to learn machine learning on your own?
**Reference (Gold):** What are the most effective self-study resources for machine learning?

---

**LLaMA2-7B (Baseline)**

How can I learn machine learning by myself, and what are the best ways?

---

**LLaDA-8B-Instruct (Teacher, 128-step)**

What are the top resources for self-teaching machine learning?

---

**OneFlowSeq (Ours, 1-step)**

What is the most effective way to self-study machine learning?

---

**Analysis** In this example, the baseline model (LLaMA2-7B) produces a grammatically correct but slightly clunky and repetitive paraphrase. In contrast, both the multi-step teacher (LLaDA-8B-Instruct) and our one-step OneFlowSeq generate highly fluent and natural-sounding questions that accurately capture the original intent. They effectively use synonyms ("top resources," "effective way," "self-teaching," "self-study"), demonstrating a deeper semantic understanding. This case highlights that our distillation process successfully transfers the teacher's nuanced generative capabilities to the one-step student model without a discernible loss in quality.

### G.2   CASE STUDY 2: QUESTION GENERATION (QUASAR-T) - ERROR ANALYSIS

This case study illustrates a typical limitation of the one-step generation approach. While often successful, the model can sometimes oversimplify or omit secondary details present in the context, a trade-off for the immense speedup.

---

**Source and Reference**

**Source Context:** Penicillin, the first true antibiotic, was discovered accidentally by Scottish physician Alexander Fleming in 1928. While studying Staphylococcus bacteria, he noticed that a mold of the *Penicillium* genus had contaminated one of his culture plates and that the bacteria surrounding the mold had been destroyed.
**Reference (Gold):** Who discovered the first antibiotic?

---

**LLaMA2-7B (Baseline)**

What did Alexander Fleming find in 1928 on a culture plate?

---

> **LLaDA-8B-Instruct (Teacher, 128-step)**
>
> Who is credited with the accidental discovery of penicillin in 1928?

> **OneFlowSeq (Ours, 1-step)**
>
> Who discovered penicillin?

**Analysis** Here, the baseline LLaMA2-7B generates a factually correct but overly specific question that misses the main point of the context (the "first antibiotic"). The multi-step teacher, LLaDA-8B-Instruct, successfully synthesizes multiple pieces of information ("accidental," "penicillin," "1928") into a comprehensive and high-quality question. Our OneFlowSeq model correctly identifies the main subject and generates a good, relevant question ("Who discovered penicillin?"). However, it fails to include the important context that it was the "first true antibiotic" or the year of discovery. This error pattern, where the model captures the primary subject but may omit secondary details in a single forward pass, represents a known trade-off of the one-step generation paradigm. While the generated question is still of high quality, it lacks the richness of the multi-step teacher's output, highlighting an area for future improvement.

## H    STATEMENT ON THE USE OF AI ASSISTANCE

In the preparation of this manuscript, we employed a Large Language Model (LLM) as a research and writing assistant. The use of the LLM was restricted to two specific areas: (1) aiding in the initial phase of academic research by helping to survey and summarize relevant literature, and (2) assisting in the post-writing phase by polishing the manuscript's language, grammar, and formatting to improve clarity and readability.

