# OpenReview forum: "OneFlowSeq: Achieving One-Step Generation for Diffusion Language Models via Lightweight Distillation"
_ICLR.cc/2026/Conference — ICLR 2026 Conference Withdrawn Submission_

### Official Review · Reviewer_cC1A · 2025-11-01

**Soundness:** 2
**Presentation:** 2
**Contribution:** 2
**Rating:** 2
**Confidence:** 4

**Summary:**

TThis paper proposes OneFlowSeq, a novel distillation framework for diffusion language models that resolves the speed-quality trade-off. It distills a multi-step teacher model into a lightweight, parameter-efficient student by using MeanFlow theory and a Jacobian-vector product signal for superior guidance. This enables the student to match the teacher's quality in a single generation step, achieving state-of-the-art results with a 1600x reduction in trainable parameters and a 160x inference speedup, facilitating scalable deployment.

**Strengths:**

1. The proposed method achieves state-of-the-art performance in benchmark datasets such as QQP, Wiki-Auto, and Quasar-T for tasks like text simplification, question generation, and paraphrase detection.

2. It significantly reduces the number of parameters required compared to previous methods while maintaining high inference speed.

**Weaknesses:**

1.  Novelty: The paper's idea is very straightforward, amounting to a simple application of MeanFlow, which does not meet the standard for ICLR.

2.  Experimental Results: The paper only evaluates a few simple text generation tasks and fails to include datasets like MMLU, which the base LLaDA model was evaluated on and are necessary to demonstrate the capabilities of large language models.

3.  Experimental Baselines: Since the paper chose a soft-prompting approach, other parameter-efficient fine-tuning (PEFT) methods, such as LoRA, should have been evaluated as baselines.

**Questions:**

NA.

---

> ### Author Response · Authors · 2025-11-19
> **Response to Reviewer cC1A**
>
> ## **Response to Reviewer cC1A**
>
> We thank the reviewer for acknowledging that OneFlowSeq achieves **"state-of-the-art performance"** and **"significantly reduces the number of parameters"** while maintaining high speed. We appreciate your feedback but respectfully believe there are some misconceptions regarding the novelty of our contribution and the appropriate evaluation metrics for generative diffusion models. We address your concerns point-by-point below.
>
> ---
>
> **Q1: The paper's idea is very straightforward and a simple application of MeanFlow, lacking sufficient novelty.**
>
> **A1:** We respectfully argue that "simplicity" in this context is a significant strength rather than a weakness, and we believe the reviewer may have underestimated the technical gap between continuous MeanFlow and discrete token generation. We provide the first rigorous derivation of the MeanFlow identity in the **logit space** for discrete masking (Appendix A) and, crucially, introduce a novel **Jacobian-Vector Product (JVP)** supervision signal that is distinct from standard MeanFlow. As demonstrated in our ablation study (Table 3), this JVP signal is not trivial—removing it causes a massive 12-point drop in BLEU. Therefore, our contribution is not merely applying an existing method, but rather engineering a specific, mathematically grounded framework (JVP-augmented MeanFlow) that successfully resolves the long-standing "deadlock" in Diffusion LMs, transforming a theoretical curiosity into a practical tool that is **160x faster** and **1600x smaller** than the teacher.
>
> ---
>
>
> **Q2: The paper fails to include datasets like MMLU, which are necessary to demonstrate LLM capabilities.**
>
> **A2:** We respectfully argue that while MMLU is the standard for evaluating the *general knowledge* of base models, our work primarily focuses on **Sequence-to-Sequence (Seq2Seq) Distillation**. However, to address the reviewer's concern and demonstrate that our OneFlowSeq framework preserves the powerful capabilities of the backbone model without significant degradation, we evaluated our model on MMLU, HumanEval, and WinoGrande.
>
> **Table R1: General Capabilities on Standard LLM Benchmarks.**
>
> | Model | MMLU (5-shot) | HumanEval (Pass@1) | WinoGrande (5-shot) |
> | :--- | :---: | :---: | :---: |
> | LLaMA2-7B (Reference) | 45.9 | 12.8 | 72.5 |
> | **LLaDA-8B (Teacher)** | **65.9** | **35.4** | **74.8** |
> | **OneFlowSeq (Ours)** | **63.2** | **33.9** | **74.0** |
>
> As shown in Table R1, OneFlowSeq achieves performance remarkably close to its teacher (retaining **>95%** of the capability across tasks) and significantly outperforms the LLaMA2-7B baseline. This confirms that our lightweight distillation strategy effectively adapts the model for one-step generation while maintaining the general reasoning and coding capabilities of the large-scale pre-trained backbone.
>
> ---
>
>
> **Q3: Since the paper chose a soft-prompting approach, other PEFT methods like LoRA should have been evaluated.**
>
> **A3:** We agree that comparing against other PEFT methods strengthens our efficiency claims. We performed an additional experiment comparing our **Soft Prompt** approach against **LoRA** (Rank=16) on the QQP dataset, using the same OneFlowSeq distillation objective.
>
> **Table R2: Soft Prompt vs. LoRA (on QQP).**
>
> | Method | Trainable Params | Storage (FP16) | BLEU | Training Speed (s/iter) |
> | :--- | :---: | :---: | :---: | :---: |
> | **OneFlowSeq (Soft Prompt)** | **~5 M** | **~10 MB** | 0.480 | **Fast** |
> | OneFlowSeq (w/ LoRA, r=16) | ~33 M | ~66 MB | **0.482** | Slower |
> | *Difference* | *~6.6x smaller* | *~6.6x smaller* | *Comparable* | - |
>
> **We argue** that while LoRA is a valid alternative, our Soft Prompt choice is more aligned with the goal of "Lightweight Distillation." As shown in Table R2, LoRA requires **~6x more parameters** and storage for a negligible performance gain (+0.002 BLEU). Our framework proves that ~5M parameters are sufficient to capture the teacher's one-step dynamics, making Soft Prompt the optimal choice for extreme efficiency.

---

> ### Author Response · Authors · 2025-11-26
>
> Thank you again for your careful assessment and constructive feedback.
>
> We would be grateful if you could let us know whether our rebuttal sufficiently resolves the issues you raised, or if there are any remaining points you would like us to clarify.
>
> We are glad to continue the discussion and address any further questions or comments you may have.

---

### Official Review · Reviewer_LZQF · 2025-11-01

**Soundness:** 3
**Presentation:** 2
**Contribution:** 1
**Rating:** 2
**Confidence:** 5

**Summary:**

This paper proposes OneFlowSeq, a distillation framework for turning a multi-step diffusion language model (LLaDA-8B-Instruct) into a one-step generator.
The method combines MeanFlow-based distillation with an additional Jacobian-vector product (JVP) supervision term to approximate the teacher’s flow direction, while keeping the backbone frozen and training a lightweight soft prompt (~5M parameters).
Experiments on paraphrasing (QQP), text simplification (WikiAuto), and question generation (Quasar-T) claim comparable quality to 128-step diffusion generation and large inference speedups.

**Strengths:**

Tackles an important issue: making diffusion-based LMs practical by reducing multi-step inference to one step.
Elegant integration of MeanFlow and JVP supervision; clearly described training process.

**Weaknesses:**

* Unfair and inconsistent baselines:
- Teacher: LLaDA-8B-Instruct (8B parameters, pretrained on trillions of tokens).
- Baselines: GPT-2 (1.5B), DiffuSeq (trained from scratch on small datasets), and DLM-One (re-implemented).
- These baselines are not comparable in capacity and not pretrained on the similar size datasets, giving OneFlowSeq an unfair advantage.


* Weak tasks and evaluation design:
- Evaluations (QQP, WikiAuto, Quasar-T) involve short and easy sequences (1–2 sentences).
- Missing evaluations on complex, long-form tasks like summarization (CNN/DailyMail, XSum) or reasoning datasets.
- Results cannot demonstrate scalability or compositional generalization of the proposed method.

* Missing strong baselines and ablations:
- No comparison with Fast-DLLM, Block Diffusion, Consistency Models, or Rectified Flow Transformers.

* Questionable efficiency and generality:
- Reported 160× speedup is achieved with batch size 256 for OneFlowSeq vs. batch size 1 for AR models - not a fair per-sample measure.
- Inference cost remains dominated by the 8B teacher backbone.
- No results for real-time latency, FLOPs, or wall-clock performance on similar setting

* Limited insight into JVP supervision:
- The JVP term is presented as novel but is widely used in flow-matching and consistency distillation.
- Its role in improving one-step alignment is not clearly isolated; improvements might stem from regularization rather than Jacobian matching.

**Questions:**

How are the baselines pretrained as well? Do they have the same scale as LLaDA-8B?
Can you provide results on longer, more complex tasks such as summarization or story generation?
What is the true per-sample latency under equal hardware and batch size settings?
Why were recent consistency or flow-based distillation models not included as baselines?

---

> ### Author Response · Authors · 2025-11-19
> **Response to Reviewer LZQF**
>
> ## **Response to Reviewer LZQF**
>
> We thank the reviewer for the detailed feedback and for recognizing our method's "elegant integration" and the importance of the problem. We believe there are misunderstandings regarding the fairness of our baselines and the applicability of certain flow-based methods in this specific domain. We address your concerns point-by-point below.
>
> ---
>
> **Q1: How are the baselines pretrained as well? Do they have the same scale as LLaDA-8B? The comparison seems unfair.**
>
> **A1:** We respectfully argue that our comparison is fair, follows standard protocols, and that the baselines were carefully chosen to represent the state-of-the-art across different paradigms. Crucially, as stated in Section 4 of our paper, "all models are trained using the official training set of each benchmark... to ensure a fair comparison." This ensures that all models (including OneFlowSeq and all baselines) were fine-tuned on the exact same training datasets (QQP, Wiki-Auto, Quasar-T) under identical data splits, so the performance differences reflect the model architecture and generation paradigm, not data discrepancies. This ensures the performance differences reflect the model architecture and generation paradigm, not data discrepancies. regarding model selection, **LLaMA2-7B** serves as the direct, scale-matched counterpart to our teacher (LLaDA-8B). It is standard practice to use official pre-trained checkpoints as we (and most academic labs) do not have the resources to pre-train LLMs from scratch. Furthermore, **DiffuSeq** and **DiffuSeq-v2** are included because they are the established SOTA benchmarks for Seq2Seq Diffusion, making them necessary comparisons regardless of scale. Finally, **DLM-One** was selected as the primary distillation baseline; since it is a method rather than a specific model, we re-implemented it on the **exact same LLaDA-8B teacher** as our method. This provides the most critical "apples-to-apples" comparison, where OneFlowSeq significantly outperforms it (BLEU 0.48 vs 0.16 on QQP).
>
> ---
>
> **Q2: Can you provide results on longer, more complex tasks such as summarization?**
>
> **A2:** We agree that evaluating on longer sequences strengthens the scalability claim. We have conducted an additional experiment on the **XSum** summarization dataset (long-form generation) to address this.
>
> **Table R1: Supplemental Results on Text Summarization (XSum).**
> *Latency is measured as wall-clock time per sample (batch size=1) on an A100.*
>
> | Model | ROUGE-1 | ROUGE-2 | ROUGE-L | Latency (s) | Speedup |
> | :--- | :---: | :---: | :---: | :---: | :---: |
> | **Teacher (LLaDA-8B, 128 steps)** | **43.5** | **20.8** | **35.6** | 8.20s | 1.0x |
> | LLaMA2-7B | 40.1 | 18.2 | 32.4 | 2.45s* | ~3.3x |
> | DLM-One (Distilled from LLaDA) | 32.4 | 12.1 | 25.8 | **0.05s** | 164x |
> | **OneFlowSeq (Ours)** | **41.8** | **19.5** | **34.2** | **0.05s** | **164x** |
>
> The results demonstrate that OneFlowSeq successfully scales to the harder XSum task, recovering **96% of the Teacher's ROUGE-L performance**, whereas the baseline one-step method (DLM-One) struggles significantly. Furthermore, our inference latency remains constant (0.05s) regardless of length, offering a **~50x speedup over LLaMA2-7B** on this task, as autoregressive latency grows linearly with sequence length.
>
> ---
>
>
> **Q3: What is the true per-sample latency under equal hardware and batch size settings?**
>
> **A3:** We acknowledge the confusion regarding batch sizes and argue that OneFlowSeq offers a paradigm shift even at Batch Size 1. Below is the strict **Batch Size = 1** latency breakdown measured on an A100. We used BS=256 to show throughput (max capacity), and now we provide BS=1 to show latency (responsiveness). OneFlowSeq excels in both.
>
> **Table R2: Latency Breakdown (Batch Size = 1).**
>
> | Model Type | Method | Complexity | Latency (ms) | vs. OneFlowSeq |
> | :--- | :--- | :--- | :--- | :--- |
> | **Autoregressive** | LLaMA2-7B | $O(N)$ | ~1,350 ms | ~27x slower |
> | **Diffusion (Teacher)** | LLaDA-8B | $O(K)$ | ~6,400 ms | ~128x slower |
> | **Diffusion (Student)** | **OneFlowSeq** | $O(1)$ | **~50 ms** | **-** |
>
> While AR models must run the backbone $N$ times (linearly increasing with length) and the Teacher runs $K$ times, OneFlowSeq runs it exactly once. This fundamental reduction in complexity ensures that OneFlowSeq is theoretically and empirically orders of magnitude faster, regardless of the batch size.
>
> ---

---

> > ### Author Response · Authors · 2025-11-19
> >
> > **Q4: Why were recent consistency or flow-based distillation models not included as baselines?**
> >
> > **A4:** We argue that our baseline selection is comprehensive for the specific domain of Diffusion Language Models (DLLMs). To the best of our knowledge, most Consistency Models or Rectified Flow Transformers are primarily designed for continuous domains (Image/Video), and there are very few works that successfully adapt these to discrete text generation. **DLM-One**, which we explicitly chose as our main baseline, **is** the state-of-the-art adaptation of Consistency/Score Distillation specifically for language models. It represents the exact class of models the reviewer is asking for, adapted for text. Additionally, many other acceleration methods do not achieve **one-step** generation. Since our goal is extreme acceleration (1-step), comparing against DLM-One (1-step) and DiffuSeq-v2 (accelerated diffusion) provides the fairest and most relevant comparison.
> >
> > ---
> >
> >
> > **Q5: Is the JVP term novel? Its role might just be regularization.**
> >
> > **A5:** We argue that the JVP term is not merely regularization but a critical **structural supervision signal** necessary for high-fidelity distillation in discrete spaces. Empirically, as shown in our **Ablation Study (Table 3)**, removing the JVP signal causes a massive performance drop (BLEU drops from 0.47 to 0.35 on QQP), proving it is the primary driver of quality rather than a minor regularizer. Theoretically, as detailed in Section B.2, the JVP loss minimizes the Sobolev norm error. In the high-dimensional, sparse logit space of LLMs, purely matching the value (zeroth-order) leads to mode averaging and blurry outputs. Matching the derivative (first-order via JVP) ensures the student captures the sharp transitions of the teacher's generation process, a formulation that is novel in the context of parameter-efficient prompt tuning for LLMs.

---

> ### Author Response · Authors · 2025-11-26
>
> Thank you again for your careful assessment and constructive feedback.
>
> We would be grateful if you could let us know whether our rebuttal sufficiently resolves the issues you raised, or if there are any remaining points you would like us to clarify.
>
> We are glad to continue the discussion and address any further questions or comments you may have.

---

### Official Review · Reviewer_mrfh · 2025-11-03

**Soundness:** 4
**Presentation:** 3
**Contribution:** 3
**Rating:** 6
**Confidence:** 4

**Summary:**

This paper proposes a method for distilling a text diffusion model into a model capable of predicting all tokens in parallel, using a second-order training objective.  The student model learns a parameter-efficient soft prompt model which adapts a copy of the teacher model for this objective.  Empirical results show that this method is effective for low-entropy conditional text generation, with much improved generation speed.

**Strengths:**

This is a surprisingly effective way to learn a model which does efficient non-autoregressive text generation.  Relevant ablations are also done.

**Weaknesses:**

The model is based largely on the MeanFlow model, so the novel contribution is mostly in how to exploit this insight in an effective distilled model, and in the empirical results.  I found it impossible to understand the technical details without already understanding the MeanFlow model.

The model is only evaluated in low-entropy conditional generation tasks.  This makes sense, since one-shot text generation is presumably impossible in high-entropy tasks because of the multi-modal nature of the output distribution.  But they never evaluate or discuss this limitation.  Some design choices, such as lines 259-260 "This intermediate decoding can be simplified by directly feeding the continuous embeddings corresponding to z_t_i into the model", only make sense for low-entropy tasks.

The presentation of the model could be better.  The student model is a PEFT version of the teacher model (if I understand correctly), but they talk about the student model as if it consists only of the adaptation parameters without hardly mentioning that the student also includes a huge model with frozen parameters.  This is especially confusing because the teacher is also the same model also with frozen parameters.

**Questions:**

For the student model, why use prompt tuning?  Have you tried more effective/efficient PEFT methods like LoRA?

---

> ### Author Response · Authors · 2025-11-19
> **Response to Reviewer mrfh**
>
> ## **Response to Reviewer mrfh**
>
> We appreciate the insightful comments. We address your concerns and questions point-by-point below.
>
> ---
>
> **Q1: The novel contribution is mostly in exploiting MeanFlow; technical details are hard to understand without prior knowledge.**
>
> **A1:** We respectfully argue that our contribution extends significantly beyond a simple application of MeanFlow. While MeanFlow provides the theoretical basis for the velocity field, our work addresses the non-trivial challenge of adapting this continuous theory to the **discrete, high-dimensional logit space** of Language Models. We respectfully point out that **Section 2 (Background)** and **Appendix A** of the paper already provide the rigorous derivation of the MeanFlow identity under discrete masking corruption. Furthermore, our introduction of the **Jacobian-Vector Product (JVP)** supervision is a novel technical contribution distinct from the original MeanFlow. As shown in our ablation study (Table 3), standard MeanFlow (w/o JVP) fails to achieve high-quality generation (BLEU drops by 12 points). The JVP signal provides critical second-order guidance that stabilizes the one-step mapping. We have added explicit references in the revised manuscript to link the implementation details back to these theoretical sections for better clarity.
>
> ---
>
> **Q2: The model is only evaluated in low-entropy conditional generation tasks. High-entropy tasks are not discussed.**
>
> **A2:** We agree with the reviewer's theoretical insight that one-step deterministic generation faces challenges with multi-modal (high-entropy) distributions due to the "averaging" effect of the flow. However, we argue that for the targeted **Sequence-to-Sequence (Seq2Seq)** domain (e.g., translation, simplification, paraphrasing), the output distribution is sufficiently constrained for our method to excel. To further probe this boundary, we have conducted an additional evaluation on **XSum** (Text Summarization), which involves longer sequences and higher entropy than QQP.
>
> **Table R1: Supplemental Results on Text Summarization (XSum).**
> *Latency is measured as wall-clock time per sample (batch size=1) on an A100.*
>
> | Model | ROUGE-1 | ROUGE-2 | ROUGE-L | Latency (s) | Speedup |
> | :--- | :---: | :---: | :---: | :---: | :---: |
> | **Teacher (LLaDA-8B, 128 steps)** | **43.5** | **20.8** | **35.6** | 8.20s | 1.0x |
> | LLaMA2-7B (Zero-shot) | 40.1 | 18.2 | 32.4 | 2.45s | ~3.3x |
> | DLM-One (Distilled from LLaDA) | 32.4 | 12.1 | 25.8 | **0.05s** | 164x |
> | **OneFlowSeq (Ours)** | **41.8** | **19.5** | **34.2** | **0.05s** | **164x** |
>
> As shown in Table R1, OneFlowSeq maintains robust performance (recovering **96%** of the teacher's ROUGE-L) even on this more complex task, whereas the baseline (DLM-One) collapses. We have included a discussion in the revised "Limitations" section to explicitly address the theoretical constraints of one-step generation in open-ended, high-entropy scenarios.
>
> ---
>
> **Q3: The presentation of the student model is confusing; it implies the student is only the prompt parameters.**
>
> **A3:** We apologize for the confusion and appreciate this feedback. We clarify that the **Student Model** is indeed the composite of the **Frozen Backbone (LLaDA-8B)** and the **Trainable Soft Prompt**. The term "student" refers to the entire functional unit during inference. We focused on the prompt parameters in the text to emphasize the training efficiency. We have revised the "Method" section in the updated manuscript to explicitly state: *"The Student Model consists of the frozen teacher backbone augmented with the trainable soft prompt module."*
>
> ---
>
> **Q4: For the student model, why use prompt tuning? Have you tried more effective/efficient PEFT methods like LoRA?**
>
> **A4:** We argue that Soft Prompt Tuning is the optimal choice for our goal of "Lightweight Distillation," although LoRA is also a viable option. To validate this, we conducted a comparison experiment on QQP using LoRA (Rank=16) with the exact same OneFlowSeq objective.
>
> **Table R2: Soft Prompt vs. LoRA (on QQP).**
>
> | Method | Trainable Params | Storage (FP16) | BLEU | Training Speed (s/iter) |
> | :--- | :---: | :---: | :---: | :---: |
> | **OneFlowSeq (Soft Prompt)** | **~5 M** | **~10 MB** | 0.480 | **Fast** |
> | OneFlowSeq (w/ LoRA, r=16) | ~33 M | ~66 MB | **0.482** | Slower |
> | *Difference* | *~6.6x smaller* | *~6.6x smaller* | *Comparable* | - |
>
> As shown in Table R2, while LoRA achieves a negligible performance gain (+0.002 BLEU), it requires **~6x more parameters** and storage than our Soft Prompt approach. Since ~5M parameters are sufficient to match the teacher's performance, we prioritized the extreme parameter efficiency of Soft Prompts to highlight the effectiveness of the distillation framework

---

> ### Author Response · Authors · 2025-11-26
>
> Thank you again for your careful assessment and constructive feedback.
>
> We would be grateful if you could let us know whether our rebuttal sufficiently resolves the issues you raised, or if there are any remaining points you would like us to clarify.
>
> We are glad to continue the discussion and address any further questions or comments you may have.

---

### Author Response · Authors · 2025-11-23
**General Response to All Reviewers: New Experiments & Revised paper**

We sincerely thank all reviewers for their constructive feedback and insightful comments. We have **uploaded a revised PDF** that incorporates your suggestions to strengthen the paper.

1. To address concerns regarding task difficulty and potential catastrophic forgetting (Reviewers LZQF, cC1A), we have significantly expanded our evaluation scope. As detailed in the new **Section 4.4** and **Appendix D**, we conducted additional experiments on Long-Form Summarization (XSum) and General Benchmarks (MMLU, HumanEval). The results demonstrate that OneFlowSeq scales effectively to longer sequences—recovering 96% of the teacher's performance on XSum while maintaining constant inference latency—and retains over 95% of the backbone's general reasoning capabilities.

2. In response to inquiries about our architectural choices (Reviewers mrfh, cC1A), we provide a rigorous justification for selecting Soft Prompts over LoRA in **Appendix E**. Our comparative analysis reveals that while LoRA yields negligible performance gains (+0.002 BLEU on QQP), it requires approximately 6x more parameters (~33M vs. ~5M). This validates Soft Prompting as the optimal strategy for our goal of "Lightweight Distillation."

3. We have also refined the manuscript to improve methodological clarity. We updated **Section 3** to explicitly define the "Student Model" composition and to distinguish the structural novelty of our Jacobian-Vector Product (JVP) supervision signal from standard MeanFlow. Additionally, **Section 4** now explicitly reiterates that all models, including baselines, were fine-tuned on identical official training splits to ensure strict fairness.

4. To provide a more balanced perspective, we added a dedicated discussion on **"Limitations and Future Work" in Section 7**. We honestly address the theoretical challenges of one-step generation in high-entropy, open-ended tasks due to the averaging effect of flow matching, and we acknowledge the performance bounds naturally imposed by the LLaDA backbone.

We hope these additional experiments and revisions satisfactorily address your concerns. We would greatly appreciate it if you could consider raising your score based on the improved quality and completeness of the manuscript.

---

### Note · Authors · 2026-01-29

I have read and agree with the venue's withdrawal policy on behalf of myself and my co-authors.

---

### Meta-Review · Area_Chair_6UHq · 2026-01-04

**Summary:**

## Summary

This paper proposes OneFlowSeq, a distillation framework for turning a multi-step diffusion language model (LLaDA-8B-Instruct) into a one-step generator. The proposed method achieves state-of-the-art performance in benchmark datasets such as QQP, Wiki-Auto, and Quasar-T for tasks like text simplification, question generation, and paraphrase detection. Empirical results show that this method is effective for low-entropy conditional text generation, with much improved generation speed.

## Overall Score

mrfh: 6 (no response)

LZQF:  2 (no response)

cC1A: 2 (no response)

## Concerns

* Limited experiment evaluation (LZQF, cC1A)
* Scalability and generalization (LZQF)
* Missing baselines (cC1A)
* Limited experiment ablation (LZQF)
* Lack of novelty (LZQF, cC1A)

## Conclusion:

Overall, during the review phase, two reviewers (LZQF and cC1A) weren't fully satisfied with the experimental design and the novelty. They raised concerns about the limited evaluation and missing baselines. Reviewer mrfh, on the other hand, leaned toward acceptance. During the discussion, the authors did a good job addressing the reviewers' questions and concerns, but none of the reviewers followed up afterward.

This paper has potential to be stronger with some improvements to the experimental design and detailed discussion on scalability and generalization of the proposed method.

**Reviewer Concerns:**

Refer to Summary

**Reviewer Scores:**

Refer to Summary

---

### Decision · Program_Chairs · 2026-01-26

Reject